# Data-centric artificial olfactory system based on the eigengraph

Seung-Hyun Sung [1,2,8], Jun Min Suh [3,4,8], Yun Ji Hwang[1], Ho Won Jang [3,5,9] ✉, Jeon Gue Park[6,7,9] ✉ & Seong Chan Jun [1,9] ✉

Recent studies of electronic nose system tend to waste significant amount of important data in odor identification. Until now, the sensitivity-oriented data composition has made it difficult to discover meaningful data to apply artificial intelligence in terms of in-depth analysis for odor attributes specifying the identities of gas molecules, ultimately resulting in hindering the advancement of the artificial olfactory technology. Here, we realize a data-centric approach to implement standardized artificial olfactory systems inspired by human olfactory mechanisms by formally defining and utilizing the concept of Eigengraph in electrochemisty. The implicit odor attributes of the eigengraphs were mathematically substantialized as the Fourier transform-based Mel-Frequency Cepstral Coefficient feature vectors. Their effectiveness and applicability in deep learning processes for gas classification have been clearly demonstrated through experiments on complex mixed gases and automobile exhaust gases. We suggest that our findings can be widely applied as source technologies to develop standardized artificial olfactory systems.

Electronicization of human senses such as artificial visual, auditory, olfactory, gustatory and tactile sensing systems has long been widely studied to replace human sensory systems[1]. In the case of artificial olfactory technology, Seiyama et al. developed the first metal oxide gas sensor using a ZnO thin film based on the redox reactions in 1962[2]. And the concept of an electronic nose mimicking the mammalian olfactory system was first proposed by Persaud et al. in 1982[3]. Since then, many researchers have developed electronic nose systems using chemiresistive gas sensors and computational analysis methods[4–7]. Recently, advanced nanotechnology (NT) and artificial intelligence (AI) are accelerating the development of bioinspired artificial olfactory systems that mimic the olfactory organs and nervous system[8,9].

However, judging from the recent research trends on artificial senses, the sense of smell has shown the slowest technological advancement among them. We point out that the reason why artificial olfactory technology has stagnated is due to the fundamental limitations on goal setting and orientation of existing researches. The main flow of discussion on gas sensors which are the basis of artificial olfactory technology has been going on with the goal for realizing high sensitivity to specific gases at low concentrations by modifying the physical and chemical properties of the nanomaterials[10–13]. The sensitivity-oriented research flows have influenced data analysis methods to demonstrate gas selectivity. In previous researches on the identification of target gas species, high sensitivity for specific gas differentiated from other gases has been regarded as high selectivity to that gas[14]. This selectivity has been verified by pattern recognition techniques such as radial fingerprint charts, contour maps, principal component analysis (PCA) or linear discriminant analysis (LDA), which

[1]School of Mechanical Engineering, Yonsei University, Seoul 03722, Republic of Korea. [2]Finance Division, Daejeon Metropolitan Office of Education, Daejeon 35239, Republic of Korea. [3]Department of Materials Science and Engineering, Research Institute of Advanced Materials, Seoul National University, Seoul 08826, Republic of Korea. [4]Department of Mechanical Engineering, Massachusetts Institute of Technology, Cambridge, MA 02139, USA. [5]Advanced Institute of Convergence Technology, Seoul National University, Suwon 16229, Republic of Korea. [6]Artificial Intelligence Laboratory, Tutorus Labs Inc., Seoul 06595, Republic of Korea. [7]Center for Educational Research, College of Education, Seoul National University, Seoul 08826, Republic of Korea. [8]These authors contributed equally: Seung-Hyun Sung, Jun Min Suh. [9]These authors jointly supervised this work: Ho Won Jang, Jeon Gue Park, Seong Chan Jun. ✉e-mail: hwjang@snu.ac.kr; jgpark@tutoruslabs.com; scj@yonsei.ac.kr

primarily rely on the sensitivity values of sensor array[7,15–19]. High sensitivity for specific gas is obviously an important factor in the development of artificial olfactory technology to overcome the low detection threshold of human olfaction. However, because there are various disturbing factors that affect the degree of the sensitivity, it is impossible to reproducibly identify the gas species using the existing methods. Even though a particular sensing material has high reactivity to a specific gas molecule, the sensitivity can vary depending on the degree of diffusion of the gas molecule reaching the gas sensor. In addition, gaseous analytes often exist as complex mixtures of individual gas molecules with varying concentrations and chemical formulas. In this case, since the overall sensitivity can be distorted due to the amplification or attenuation of the reactivity by the competition between individual gas molecules in the mixed gas, it is difficult to expect accurate gas identification results with the sensitivity-oriented data analysis methods[5,20,21].

On the other hand, there is high probability that the wasted data except for sensitivity contains significant amounts of important information for specifying the gas identities. Therefore, in order to overcome the conventional issues derived from sensitivity-driven data composition which is vulnerable to the environment changes, it is necessary to newly define the meaningful data characteristics required for gas identification and develop an appropriate signal processing and interpretation technique to effectively characterize them. To realize this, the uniqueness, veracity, stability and reproducibility of time series data waveforms according to the types of reactive gases and sensing materials should be preferentially considered in sensor development. In other words, it is necessary to develop standardized gas sensors which generate eigengraphs which have highly refined response waveforms by optimizing the redox reactions of sensing materials to target gases, just as humans perceive a specific gas molecule as a somewhat common smell regardless of racial, cultural and national background. Then, discriminative and robust feature vectors that effectively reflects the hidden attributes of odors should be extracted from the eigengraphs for universal and reproducible identification.

However, until now, there has been little interest in the above data-centric approach in the entire process from sensor development to data analysis. In this study, we proposed a sophisticated artificial olfactory system inspired by human olfactory mechanism. This system consists of a sensor array that mimics the physiological functions of human olfactory receptors and an AI analysis process that reflects the olfactory perception properties of the cerebral limbic system. These processes include optimization and standardization of nanomaterials, dedicated measurement equipment for sensor arrays, mathematical feature engineering and deep learning analysis. We suggest that the data-centric approach can be a promising solution to develop standardized artificial olfactory system that can perfectly identify delicate differences in various odors composed of complex mixture gas molecules.

## Results

### Artificial olfactory system inspired by the human olfactory mechanism

In 1991, a breakthrough research on mammalian olfactory receptors by Richard Axel and Linda Buck contributed to the establishment of a physiological and medical basis for the olfactory mechanism[22,23]. Specifically, it was discovered that humans have about 350 types of olfactory receptors, and each olfactory receptor is activated by chemically reacting with the specific volatile gas molecules entering the nose[24]. The encoded family of the activated olfactory receptors generates and transmits unique electrical signals, which are transmitted through the olfactory tract and synthesized in the olfactory region of the cerebrum and recognized as a certain smell. Our artificial olfactory system has various functional similarities because it was developed with inspiration of the olfactory mechanism of the human body. The

components of this system corresponding to each olfactory organ of the human body are summarized as follows (Fig. 1).

First, an olfactory receptor-like sensor array (ORSA) which generates various eigengraphs depending on the type of redox reactions between gas molecules and sensing materials was invented to mimic the physiological function of olfactory receptors. In this process, a top-down deposition technique using the electron beam evaporator with glancing angle deposition (GLAD) function was utilized[25–27]. Two different types of surface functionalization engineering including noble metal nanocatalyst decoration and metal oxide nano-heterojunction were simultaneously applied to impart different gas sensing characteristics to each channel of the ORSA[28–30]. In addition, chemical deterioration and changes in physical properties due to repeated measurement over a long period of time have been minimized by optimizing the manufacturing conditions of the sensing materials.

Second, the measurement and monitoring system inspired by the role of the olfactory bulb and olfactory tract serve as a pathway for relaying and transmitting the generated signals. In order to maximally extract the intrinsic characteristics of the redox reactions, we utilized a dedicated measuring instrument that can assist the gas reactions in occurring under stable conditions by minimizing external environmental factors such as humidity, temperature and vibration.

Third, we designed an AI analysis process to identify odors by mimicking the olfactory perception properties of the limbic system including the piriform cortex, amygdala and hippocampus. The piriform cortex and amygdala primarily receive odor information converged from the glomeruli of the olfactory bulb and encode them into unique odor representations along with associated episodic memories to specify individual odor identities[31,32]. The hippocampus can learn odor representations and store them in the form of long-term memories, so that when a random odor is input, the associated memory can be immediately recalled to recognize the identity of odor[33,34]. These process were respectively imitated by feature engineering and deep learning analysis. In the feature engineering stage, the odor attributes contained in the eigengraphs were extracted and mathematically represented as the Mel-Frequency Cepstral Coefficient (MFCC) feature vectors based on the fast Fourier transform (FFT). The Fourier transformation is mathematically expressed as an infinite series of products of the complex exponential functions and their weights (magnitudes of periodic function), which can be decomposed into the complex number of the sinusoidal periodic by Euler's formula[35]. That is, the key value of utilizing the Fourier transform is that any time-domain signal can be transformed into the frequency-domain components which represent as a unique linear combination of frequency components consisting of sine and cosine functions with various magnitude and phase. Accordingly, Fourier transform has been widely applied in various fields requiring signal processing such as communication engineering, vibration analysis, and speech recognition[36,37]. Therefore, the Fourier transform can be an attractive method to explore the potential intrinsic attributes of gas sensing properties from eigengraphs with various waveforms generated by the unique redox reactions between sensing materials and gas molecules. MFCC are feature vectors developed by imitating the human audible frequency band caused by the nonlinear structure of the cochlea to characterize the speech signal, which can be used as a dimensionality reduction method that selects only core features representing eigengraphs from Fourier transformed frequency components. Finally, in the deep learning stage, the MFCC feature vectors were input into the deep neural network model which learned them to classify the identities of mixed gas molecules.

### Design strategy and fabrication of olfactory receptor like sensor array

Just as the human body's olfactory receptors generate unique electrical signals, our research begins with the hypothesis that different

types of sensing materials will have unique reaction properties for specific gas molecules. It is reasonable to assume that there are differences in the degree of redox reactions because the physical and chemical properties of each gas molecule are different due to differences in component elements, bonding methods and bonding structures. We devised a systematic and efficient fabrication strategy to impart different gas sensing characteristics to each channel of the sensor array through a sequential top-down deposition process. Since the top-down deposition technique provides thin films with uniform quality over a large area, it is advantageous for securing the normality, reproducibility and stability of the sensor signal and potentially designing a stable international standard detector such as human olfactory receptors[27]. Here, we introduce a human-inspired design strategy to fabricate the ORSA combinatorially appling noble metal nanocatalyst decoration and metal oxide nano-heterojunction techniques[38,39], which are known as representative methods for the development of high-performance gas sensor through physical vapor deposition[28–30].

The vertically aligned $SnO_2$ nanorods selected as the main reaction source to gas molecules are one of the nanostructures that can be easily fabricated through the GLAD without a complicated synthesis process. The porous film with nanorods exhibits superior gas sensing properties because it provides a larger adsorption area for gas molecules compared to the thin film structure. The process is summarized as follows (Fig. 2a). At the beginning of deposition, the vapor flux evaporated by the electron beam reaches the substrate and forms a number of randomly dispersed nuclei. As the nuclei gradually grow, self-shadow regions are formed in the blocked area where additional vapor flux cannot reach due to the height of the nuclei, and nanorods grow in an inclined direction. Additionally, when rotation is applied in a tilted state, the nanorods are aligned in a perpendicular direction to the substrate. During subsequent heat treatment, the deposited bimetallic nanolayers of transition metals (Co, Ni, Cu) and noble metals

(Pt, Pd, Au) agglomerate together and are decorated in the form of nanoparticles on the $SnO_2$ nanorods. At the same time, the transition metals were oxidized to p-type transition metal oxides ($Co_3O_4$, NiO, CuO). In addition, based on the above design strategy, we have developed a wafer-scale manufacturing process compatible with the semiconductor process for fabricating highly integrated sensor array chips (Fig. 2b). The deposition process of each material was sequentially performed with the assist of spatially addressable shadow masks that accurately expose only the target area in a specific column and row direction. Finally, the fabrication was completed by cutting the wafer substrate into single chips.

We confirmed that all sensing channels of the ORSA were successfully deposited. Cross-sectional and top-view scanning electron microscopy images of nanostructures deposited on the nine sensing channels of the 3 × 3 ORSA were shown in the Supplementary Fig. 1. As shown in Supplementary Fig. 1a–c, the aggregation phenomenon of the bimetallic nanolayers in the form of nanospheres on the upper part of the nanorods significantly occurred when Au was included. Actually, The X-ray diffraction peaks of Au component were noticeably observed due to the agglomeration of Au. It was demonstrated that Au nanoparticle was polycrystalline with the (200) and (311) preferred orientations by indexing the JCPDS# 04-0784 (Supplementary Fig. 2). Exceptionally, despite the small degree of agglomeration of Pt (Supplementary Fig. 3b, c), the XRD peak (111) of a single crystal Pt nanoparticle was weakly observed by indexing the JCPDS# 04-0802. On the other hands, it was presumed that the XRD peaks of Pd were overlapped with the $SnO_2$ peaks due to small crystallization and not observed. The CuO, NiO and $Co_3O_4$ oxidized from the transition metals (Co, Ni, Cu) deposited under the noble metals (Pt, Pd, Au) were not observed on the surface through XRD. As an applicable future perspective inspired by this study in terms of single gas sensor research, our design and manufacturing strategy will contribute to overcoming material wastage and time-consuming trial and error in the process of

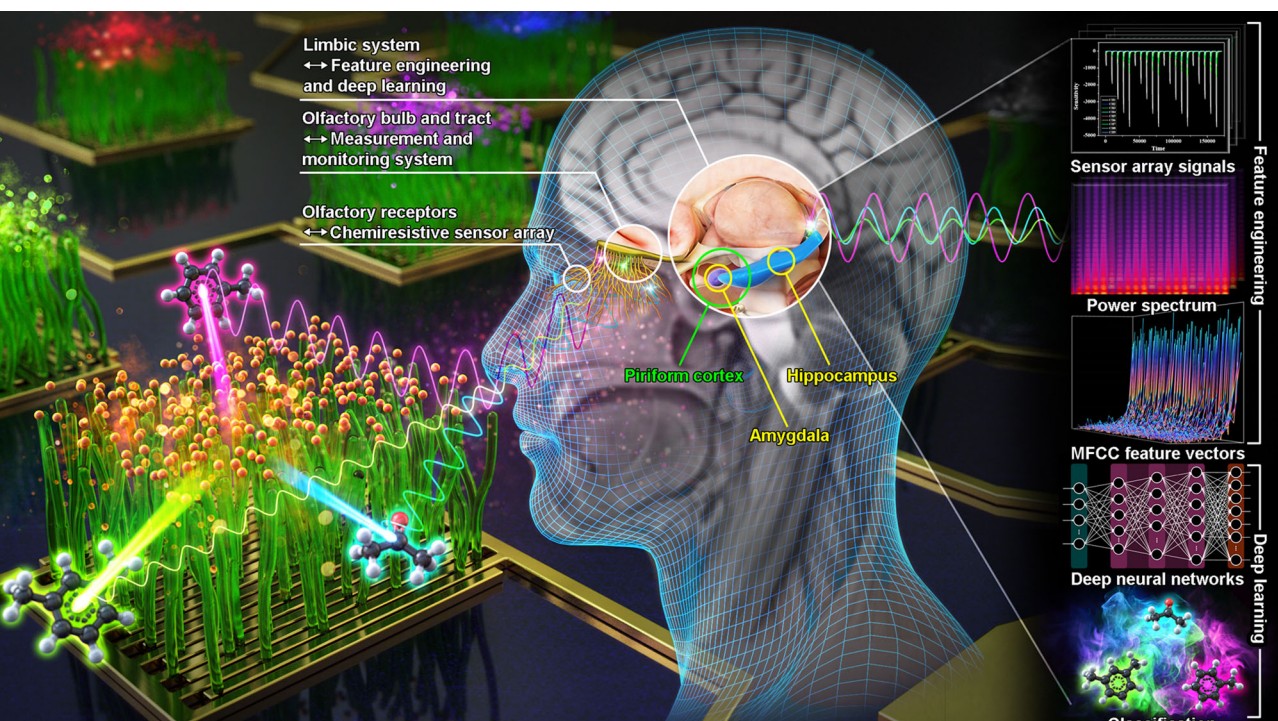

**Fig. 1 | Conceptual schematic of the artificial olfactory system inspired by the human olfactory mechanism.** The chemiresistive sensor array mimicking the olfactory receptors chemically reacts with gas molecules to generate different electrical signals with unique waveforms. The measurement and monitoring system serve as the function of olfactory bulb and tract which separately relay and transmit the signals to the limbic system. The mathematical feature engineering and deep learning analysis mimic the human olfactory perception properties of limbic system in the cerebrum such as piriform cortex, amygdala and hippocampus.

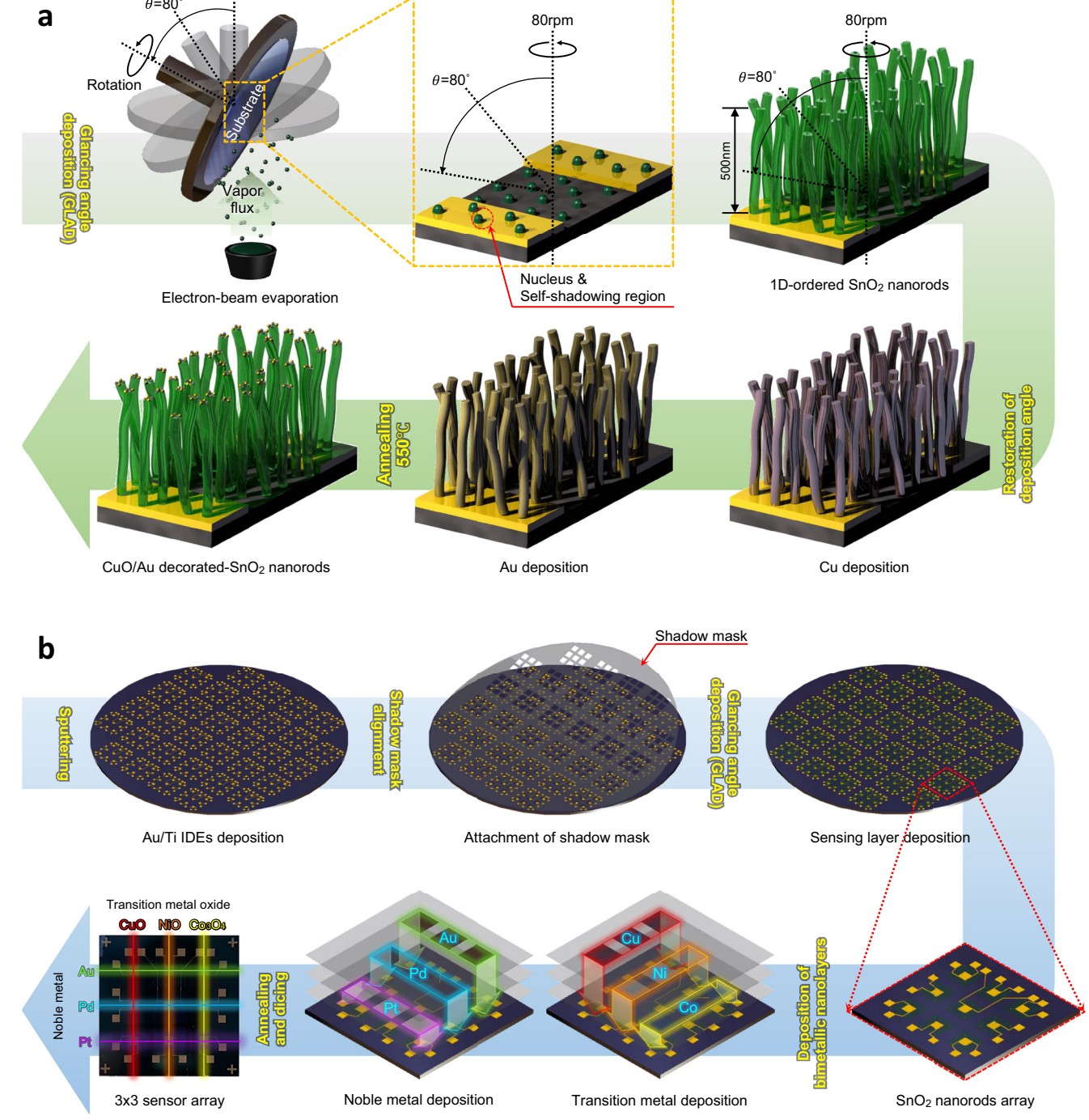

**Fig. 2 | Schematics of the top-down deposition process to fabricate olfactory receptor-like sensor array. a** Combinatorial design strategy of individual sensor based on the SnO₂ nanorods functionalized with composite nanocatalyst composed of noble metal (Au) and transition metal oxide (CuO) using GLAD of electron beam evaporation. **b** Wafer-scale mass production through semiconductor process for depositing SnO₂ nanorods and bimetallic catalysts including noble metals (Pt, Pd, Au) and transition metals (Co, Ni, Cu) assisted by the spatially addressable shadow masks.

discovering and optimizing the optimal combination of sensing materials, which will be expected to be advantageous for commercialization through mass production.

### Optimization of the deposition thickness of bimetallic nanolayers

In order to generate optimal reaction waveforms that satisfy the above mentioned four conditions, it was necessary to optimize the deposition conditions of the bimetallic nanolayers. In this process, we focused on elucidating the role of the composite nanocatalyst

composed of noble metal and transition metal oxide. Basically, it is assumed that the gas sensing signal is affected by the electronic properties modified by the catalytic nanojunctions. The p-n heterojunction between the p-type transition metal oxide nanoparticles and the n-type SnO₂ nanorods transforms the electrical signal by inducing interfacial electron transfer due to the band gap difference. Noble metal nanoparticles act as electronic sensitizers to improve gas sensing performance due to the spillover effect[40]. Additionally, the magnitude of the initial baseline resistance and sensitivity are changed depending on the type and amount of the nanocatalyst[29]. Accordingly,

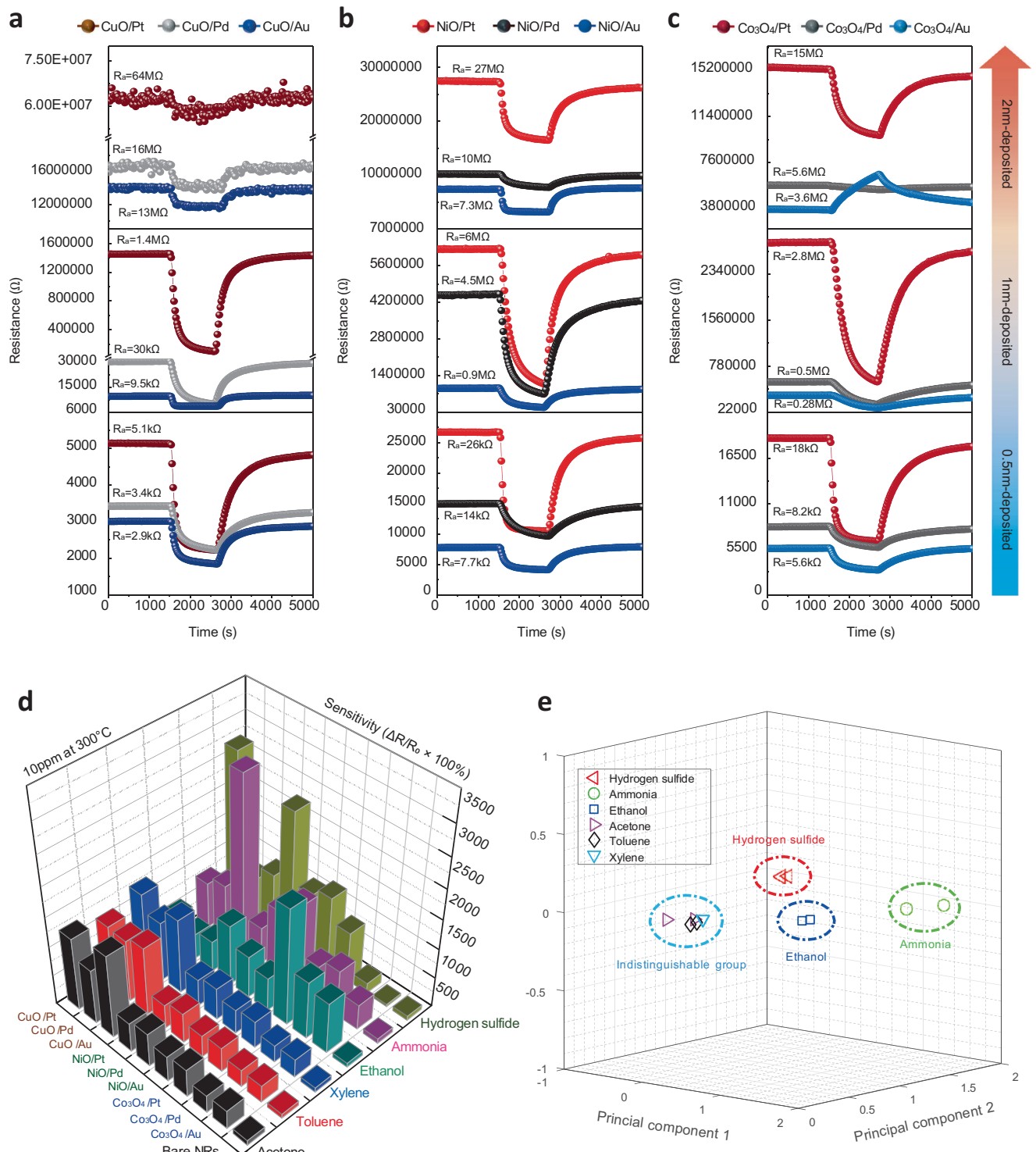

**Fig. 3 | Investigation of gas sensing characteristics of olfactory receptors-like sensor array to various volatile organic compounds. a–c** Change of initial baseline resistance according to the individual deposition thickness of bimetallic nanolayers. **d** Sensitivity values of 1nm-deposited sensor array to the 6 gas species (10 ppm of acetone, toluene, xylene, ethanol, ammonia and hydrogen sulfide) at the 300 °C. **e** PCA result based on the sensitivity values of all channels of the optimized sensor array.

the reactivity to 10ppm acetone was evaluated conditions by changing the deposition thickness of each metallic nanolayer to 0.5 nm, 1 nm, and 2 nm, respectively (Fig. 3a–c).

First, it can be seen that each channel generates unique reaction waveforms according to the combination of the catalytic materials. Also, each signal generally showed a reduction reaction behavior of n-type metal oxide. Exceptionally, the 2 nm-deposited $Co_3O_4$/Au

channel exhibited an opposite reaction pathway such as the p-type metal oxide behavior to the reducing gas. As the thickness of the transition metal deposited on the $SnO_2$ nanorods gradually increases, the active surface area of $SnO_2$ for gas reaction decreases and may eventually be blocked[41]. As a result, the p-type transition metal oxide can act as a main sensing source. In this process, the electrical stability and sensitivity of the sensor may be deteriorated due to the

continuous reaction competition between the p-type metal oxide and the n-type SnO2. Moreover, the p-n heterojunction causes the recombination of electrons and holes pair, which decreases the domain carrier density on the SnO2 surface. This can lead to an increase in the baseline resistance, which can be further promoted as the amount of the p-type transition metal oxide increases[41,42]. Actually, it was confirmed that the initial baseline resistance tended to increase as the total deposition thickness of the bimetallic nanolayers increased. The 2 nm-deposited samples formed high resistance signals that were difficult to measure through general electrical measuring equipment. In particular, it was confirmed that unstable signals with noises were generated from the 2 nm-deposited CuO-based samples. On the other hand, the 0.5 nm-deposited samples were electrically stable due to a relatively small deposition amount. However, since the basic resistance was low, it was difficult to expect visible changes in electrical properties by the catalytic nanojunctions. Compared with the 1nm-deposited sample, the diversification of the reaction waveforms according to the type of the composite nanocatalyst was not significant, and the gas sensing performance was not relatively improved. As a result, we optimized our fabrication specifications by setting the thickness of each metallic nanolayer to 1 nm.

Meanwhile, the magnitude of baseline resistance in all channels was confirmed in the order of Pt > Pd > Au according to the type of noble metal, regardless of the deposition thickness. The degree of change in the electronic properties by the catalytic nanojunctions can depend on the surface contact area and interface length between the nanocatalyst and the support[29]. So, we noted the morphological changes of the bimetallic nanolayers during the annealing process. As shown in Supplementary Fig. 1a-c and 3f, g, it was observed that the agglomeration phenomenon of the bimetallic nanolayers in the form of nanospheres on the upper of the nanorods occurred significantly when Au component was included. However, in the absence of Au component, no noticeable morphological changes such as agglomeration were observed after heat treatment as shown in Supplementary Fig. 1d–i and 3b–e. The channels containing Pd and Pt components with the small degree of agglomeration had the large surface contact area and long interface length of nanocatalyts, so the degree of change in resistance and sensitivity by the catalytic nanojunctions was relatively large (Fig. 3). Actually, the electron transfer between the nanocatalysts and SnO2 nanorods was confirmed by XPS analysis. The Sn 3d$_{5/2}$ peaks of the NiO-based channels were calibrated based on the C 1 s peak (284.8 eV) and compared to the bare SnO2 (486.6 eV). In the Pt and Pd cases, the binding energies of the Sn 3d$_{5/2}$ peak was shifted higher values (486.9 eV), which means that the electron transferred from the SnO2 to the nanocatalysts to increase the baseline resistance. On the other hands, the Sn 3d$_{5/2}$ peak of the Au was shifted lower values (486.5 eV), which means that the electron transferred from the nanocatalysts to the SnO2 to decrease the baseline resistance (Supplementary Fig. 3a). That is, it was found that the difference in the degree of agglomeration according to the amount and type of noble metal affected the gas sensing performance. Eventually, we confirmed that the reaction pathway and waveform of gas sensing signal could be modified by precisely controlling the amount, type, morphology and distribution of the catalytic nanojunctions to satisfy the four major conditions of eigengraph.

### Investigation of the preliminary gas selectivity using principal component analysis

To confirm the sensitivity-based preliminary selectivity of the optimized 1nm-deposited ORSA, we investigated the gas sensing performance for various volatile organic compounds (VOCs). We also compared the gas sensing performance of the ORSA with a bare SnO2 nanorods gas sensor to confirm the effect of composite nanocatalyst functionalization. The sensitivity was calculated as S = [(R$_{air}$ − R$_{gas}$) / R$_{gas}$] × 100% from the reaction graph for 10 ppm of acetone

(CH3COCH3), xylene (C6H4(CH3)2), toluene (C6H5CH3), ethanol (C2H6O), hydrogen sulfide (H2S) and ammonia (NH3) gases. R$_{air}$ is the basic resistance to fully saturated dry air and R$_{gas}$ is the gas resistance at the end of the gas reaction. The bare SnO2 nanorods sensor mostly exhibited similar sensitivities in the 70–80% range for each gas molecule. On the other hand, each sensor of the ORSA significantly improved the sensitivity compared with the bare SnO2 nanorods sensor (Fig. 3d). In particular, the sensitivity of the SnO2-CuO/Au sensor to 10ppm ethanol and NH3 was increased by 27.3 and 37.15 times than the bare SnO2 nanorods sensor, respectively. In addition, the selectivity of each channel for a specific gas was improved according to the combination of catalytic materials, which contributed to the identification of gas molecular species by generating various response patterns of the ORSA. The PCA result based on the sensitivity values of all channels of the sensor array showed that plots representing ethanol, ammonia and hydrogen sulfide were well distinguished from each other. Nevertheless, the response patterns to acetone, toluene, and xylene gases, which are known to have low reactivity due to strong carbon-hydrogen bonds and stable benzene ring structures, were similar each other[43,44]. Therefore, the plots of acetone, toluene and xylene overlapped each other, making it difficult to distinguish them (Fig. 3e). To solve this problem, deep learning analysis was performed on the three indistinguishable VOC gases through PCA. The preparation of a database for the followed deep learning analysis was described in Methods.

### The law of existence of Eigengraph in electrochemistry and its implementation conditions

In this section, formal definition of the eigengraph and the conditions for generating eigengraphs are described based on the phenomenological observation of our experimental result of gas reaction using metal oxide-based gas sensors. Since the gas response signal of gas sensor is the result of the interaction between the sensing material and the gas molecules, the justification for the existence of the eigengraph is explained by considering all the theoretical factors for each.

First, in terms of gas sensors, the receptor function and the transducer function are the key factors explaining the principle of the gas detection mechanism and the generation of the detection signal (as a result of the reaction)[45–47]. Receptor function is related to the chemisorption, desorption and redox reaction of gas molecules on the surface of the sensing material. The transducer function converts the results of chemical interaction with gas molecules into electrical signals which flow through the conduction channels with the Schottky-barriers formed by interconnection of the grain boundaries of adjacent nanoparticles. The surface charge density of sensing materials is changed by the redox reaction that occurs during the chemisorption of gas molecules, resulting in the formation of a charge depletion region. This leads to the change of the height of the Schottky barrier, which changes the resistance of the conduction channel[47,48]. In other words, the receptor function and the transducer function are closely related to each other, and these surface reaction properties basically depend on the intrinsic physical and chemical properties of the elements constituting the sensing material, and can also be influenced by manufacturing and synthesis methods. For example, electrical properties of semiconductor materials, such as density of states, energy band structure, and carrier concentration, are unique depending on the type of constituent elements. In particular, when metal or metal oxide nanocatalysts having different Fermi levels are added, electrical properties of the sensing material are modified by carrier diffusion in the junction region[49,50]. In addition, the specific surface area for chemisorption of gas molecules can be improved through the geometric/morphological modification of the nanostructure according to the manufacturing method, and the electronic conduction can be improved by controlling the crystallinity to induce grain size effect[50–52].

Second, in terms of gas molecules, the theoretically established power law for the response of metal oxide semiconductor-based gas

sensors can be summarized as a function of gas partial pressure and sensor resistance, by combining the acceptor function and the transducer function[53,54]. It means that the power law can be influenced not only by the type of gas molecules but also by the above-mentioned material factors affecting the surface reaction properties. Judging from this, it can be reasonably inferred that the electronegativity according to the type of element constituting the gas molecule, the type of chemical bonding such as covalent bond and hydrogen bond, and the bonding energy according to the molecular structure can also affect the interaction with the sensing material. Consequently, it is obvious that the eigengraph is generated as a result of interaction between a specific sensing material and gas molecules.

However, since the existing theories do not consider the concept of time change, so there is a limit to predicting the actual change in sensor resistance over time. Until now, there have been few reports of systematic experimental results on signal formation for long-time reactions that support widely established theoretical explanation. As a result, waveforms in the generated graphs have not been considered as important factors. Therefore, based on the major four conditions for time series gas response waveforms described in the introduction section, we focused on finding optimized signals with refined waveforms by controlling our advanced nanotechnology and measurement environments. Eventually, we experimentally found that different intrinsic signals can be created by the redox reactions between specific sensing materials and gas molecule using an optimized sensor array which have 9 independent and stable gas sensing characteristics. Thus, we formalize the findings as the law of existence of Eigengraph in electrochemistry, which is intended to represent the existence of natural intrinsic electrochemical reactions among the nanomaterials. In order for the eigengraphs to be commonly used with engineering credibility, researchers must create an internationally standardized and optimized gas sensors and appropriate peripheral hardware that ensures reliability in various measurement environments. Based on this, it is necessary to find and collect the eigengraphs that satisfy the major four conditions of time series gas response data for various gas molecules under various environmental conditions to identify unknown odors. The law of existence of Eigengraph in electrochemistry can be applicable to all fields dealing with various electrochemical reactions and provide potential opportunities to inherently understand their unique characteristics by extracting significant attributes from eigengraphs. Consequently, it will ultimately be possible to predict unknown target molecules through the eigengraphs for well-established reactions based on advanced nanotechnology.

## Deep learning based on the eigengraph analysis for identification of gas types

Existing analysis methods for gas selectivity, which did not consider the gas sensing signal waveform, have limitations in distinguishing gas types using similar magnitude-based input variables such as sensitivity, response and recovery times[18,21]. Apart from the conventional well-known gas sensing characteristics, it is necessary to develop a dedicated signal processing system to extract the hidden intrinsic attributes from the eigengraph and characterize them as decisive features. In this study, the MFCC feature vectors based on the Fourier transform were introduced as input data for deep learning model to effectively express the reaction waveform between gas molecules and nanomaterials. According to the Fourier transform theory, even any signals with similar amplitudes can be decomposed by combining different fundamental and harmonic frequencies at a unique ratio, so it can be expected to overcome the selectivity confusion caused by the cross-sensitivity of the sensor array[35]. The FFT algorithm was utilized to rapidly perform Discrete Fourier Transform (DFT) for the generation of spectrum, and it has been applied to discrete points constituting the raw signal in the time domain[55,56]. The unknown eigencomponents constituting the time series raw signal were converted into frequency

components with various amplitudes and phases, and their band-specific intensity corresponding to each frequency bin was shown in the FFT spectrum[57]. Supplementary Fig. 5 shows the estimation of normalized power spectral density of squared FFT results assigned to the linear frequency bins[56]. Although the FFT spectrum itself reflects the strong uniqueness of the raw signal, it is necessary to further reduce the dimensionality of the input space to implement efficient data processing flows in deep learning networks by minimizing computational load such as data processing and training times[58]. Therefore, it is necessary to select and intensively investigate the most prominent features among numerous variables.

MFCC is the result of imitating human audible frequency resolution by converting linear frequencies to logarithmic low frequencies, which can prevent training overfitting of deep learning networks and improve recognition accuracy by densely analyzing only key attributes of the eigengraph[59,60]. The MFCC feature vectors were obtained by using librosa package of the Python developed for acoustic and audio signal processing[61]. Figure 4a shows the summary of MFCC feature vectors extraction flow consisting of three steps: (i) preprocessing of time series raw signals (ii) generation of frequency domain power spectrum (iii) extraction of 20-dimensional MFCC feature vectors. The first order MFCC components which was negative number and has large deviation was excluded from the final diagram of the Fig. 4a. The linear frequency power spectrum with the mean squared magnitudes of the FFT results was converted to the Mel-spectrogram with non-linear Mel-scale bins by passing through the Mel-filter banks[62]. Thereafter, the log Mel-spectrogram reflecting the human hearing characteristics was generated by logarithmic compression[57], and the 20-dimensional MFCC feature vectors were finally obtained through inverse discrete Fourier transform (IDFT).

As shown in the Fig. 4b, the deep neural network consisting of three hidden layers between the input and output layers was devised to recognize the gas reaction eigengraphs. The output goal of the deep learning architecture using the 20-dimensional MFCC feature vectors as input data was classification of all 117 classes labeled depending on the gas species and mixing ratio. An initial training of the deep learning model was performed by using full signals of 342 points. Additionally, in order to improve the computational efficiency required for training the deep learning model by reducing training sample size, we progressively investigated the classification performance using top 171 and 50 points samples as training sample and compared them with the result using the full signals. Since the 50 points section is an important part where the target gas molecules and the sensing materials chemiresistively react each other, so it is considered as the minimum signal unit containing only pure eigencomponents. In other words, by using the pure gas response as input data, we can verify how efficiently the MFCC feature vectors represent the intrinsic attributes of the eigengraphs and contribute to the classification of gas species.

At the same time, in order to verify the learning efficiency of the devised deep neural network for MFCC feature vectors, the learning rate was adjusted to investigate the effect of the learning rate on the model performance. The learning rate is a parameter that controls the degree to which weights between nodes of a neural network are updated through iterative learning on input data[63]. In other words, it is possible to find out how quickly the deep learning model can discriminately learn and adapt to input variables. When using a small learning rate, it can usually take a long time to converge to the target value due to the slow learning process. On the other hand, when a large learning rate is used, the optimal value required for weight update can be overlooked due to the large step size of the optimization algorithm which derives the minimum of the loss function, resulting in an unstable learning process[63]. Considering the effect of the learning rate on the model performance, we investigated the training accuracy and loss for the three training sample sizes using learning rates of 0.00001 and 0.0001 and presented them in Fig. 4d, e. The training accuracy for

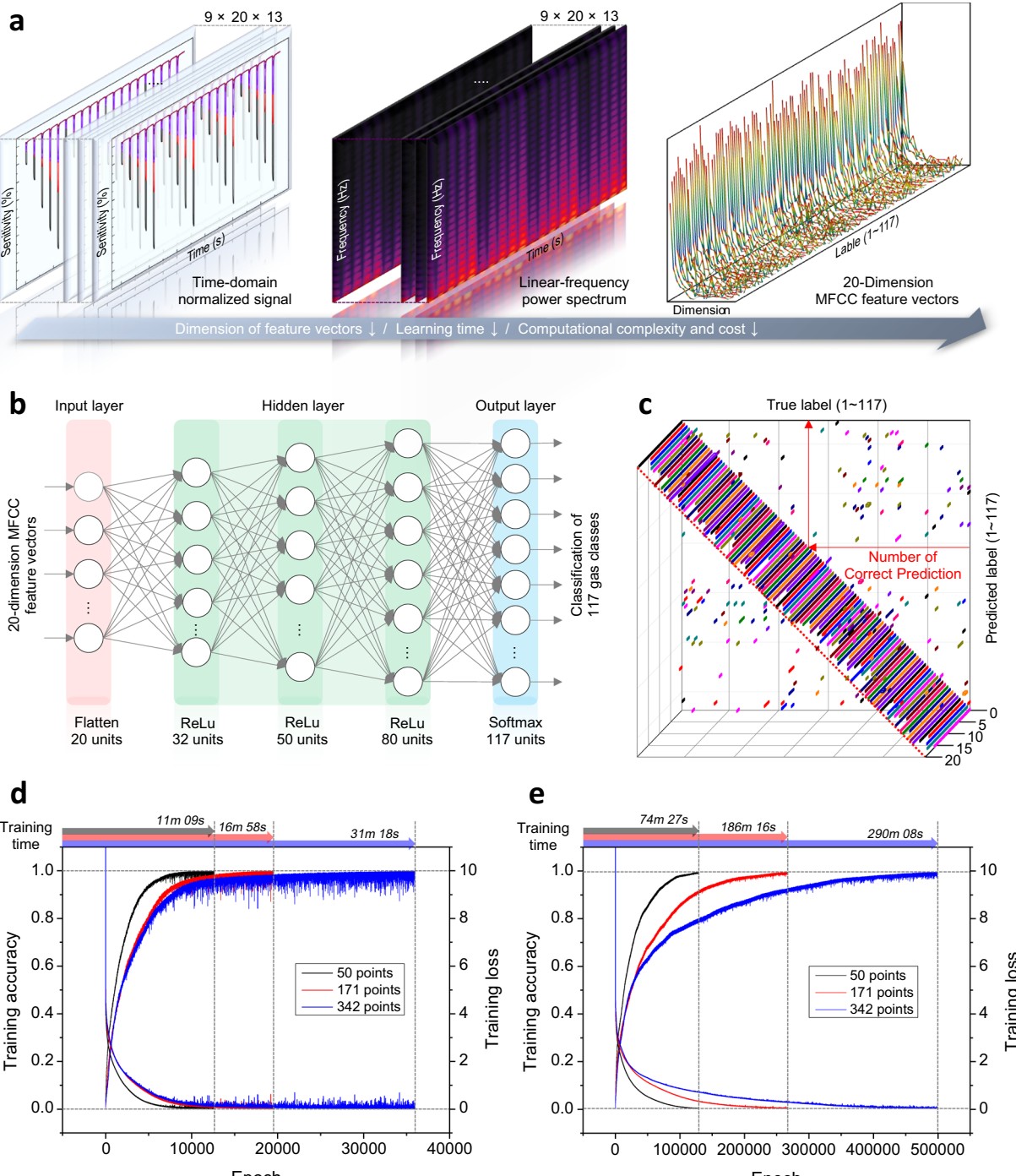

**Fig. 4 | Eigengraph deep learning analysis process. a** Preprocessing flow for extraction of 20-dimensional MFCC feature vectors based on fast Fourier transform results. **b** Deep neural network with 3 hidden layers activated with ReLU function. **c** 3D confusion matrix indicates the number of correct prediction to confirm confusable sets between the 117 predicted and true labels when the training accuracy was temporarily 90%. **d**, **e** Training accuracy and training loss graphs according to the training sample size using learning rate of 0.0001 and 0.00001, respectively.

each was set at 99.9% and 99.5% and training loss was both set to zero. Interestingly, it was confirmed that the training accuracy and loss completely and monotonously converged to the target value in all three cases regardless of the learning rate. Thus, the findings were derived that MFCC is a promising feature vector that makes discrimination between the 117 labeled classes very effective. Since the training accuracy fully saturated to the target values for all cases, subsequent discriminative training was not required. Alternatively, in order to identify the confusable sets during the model training using

the 50 points samples and 0.00001 learning rate, the 3D confusion matrix comparing predicted labels with true labels when the training accuracy reaches 90% was represented in Fig. 4c.

But there was a difference in the time to reach the target value. For the three cases in Fig.4d, the number of iterations required for 99.9% saturation of training accuracy was 12,635, 19,485 and 35,947 epochs for the 50, 171 and 342 points. In proportion to this, the training time gradually decreased to 31, 16, and 11 min as the training sample size decreased. For the three cases in Fig. 4e, the number of iterations

required for 99.5% saturation of training accuracy was 129,274, 266,754 and 499,029 epochs for the 50, 171 and 342 points. Similarly, the training time gradually decreased to 290, 186, and 74 min as the training sample size decreased (Supplementary Table 5.). This suggests that the signals enlarged by unnecessary information of the recovery stage made it difficult and complicated to determine the hyper-plane dividing the feature pattern space, resulting in increasing the computational complexity and learning time. On the other hand, it can be seen that the 50 points sample was enough to train model, and contained only determinant attributes for the gas reaction, so it can be orthogonally separated by simple learning. As a result, although the training sample size was reduced by about 85.4% of full signal, the computation time was highly saved up to 3–4 times as well as maintaining 99.9% accuracy. This is contrary to the general tendency that the deep learning accuracy decreases as the training sample size is reduced[58]. This could be reasonably expected because the input space was kept invariability due to the optimized unique waveforms generated from the optimized sensing materials.

Additionally, Euclidean distance was calculated to analyze the uniformity and reproducibility between the eigengraph waveforms of each channel of the ORSA for the target gas (Supplementary Table 6). It can be seen that the Euclidean distance value of the main diagonal component is much smaller compared to the values calculated between different channels, which indicates that the optimized reaction characteristics between the same sensing material and gas molecules were stably maintained, thereby generating reproducible sensing signals. On the other hand, since the eigengraphs between different channels are unrelated, the Euclidean distance values became larger.

Consequently, the functional effectiveness of the dimensionally reduced MFCC feature vectors for the pure gas response was obviously demonstrated through downsizing the training sample size and controlling the learning rates. This suggests that the strategy for designing data-centric AI system by an optimized high quality data generation process for extraction of MFCC feature vectors can be a promising approach for the realization of generalized artificial olfactory technology.

**Empirical experiments on automobile exhaust gases**

Through the previous sections, we have sufficiently demonstrated the utility and suitability of MFCC feature vectors for characterizing the intrinsic attributes of the eigengraphs waveforms. In this section, we confirm the potential of our artificial olfactory system for a real-life engineering application models through empirical experiments on exhaust gases and their individual components depending on the type of automobile engines. Recently, as the negative effects of automobile exhaust gases on air pollution and human health have been revealed, many policy-level and engineering efforts such as establishment of related laws and development of various emission-reducing technologies are continuing to reduce exhaust gas emissions around the world. In particular, the diesel exhaust gas contains lots of nitrogen oxides and particulate matters known as class 1 carcinogen, which can cause very harmful cancers and respiratory diseases in the human body and photochemical smog and acid rain in the atmosphere[64]. Therefore, it is important to establish enhanced standards and impose regulations for vehicles equipped with specific engines to reduce air pollution, which can be achieved by accurately identifying the critical factors according to the engine types adversely affecting human health and air pollution in urban areas caused by the exhaust gases. However, automobile exhaust gases typically contain a mixture of various types of oxidizing and reducing gas species such as hydrocarbons, nitrogen oxides ($NO_x$), and carbon oxides, both reactions can occur, it is difficult to predict the gas reaction characteristics of such mixed gases through existing theories that primarily concern the reaction between a single type of gas and a reactant. To the best of our knowledge, any study has not yet been reported on elucidating the gas reaction characteristics of complex unknown mixture gases containing both oxidizing and reducing gases by considering their eigengraph waveforms. Besides, analyzing individual components of exhaust gas emitted in trace amounts requires applying an appropriate analysis method to each gas, which makes comprehensive analysis difficult and takes a lot of time and money. For example, Orsat gas analysis, gas chromatography, infrared spectroscopy and non-dispersive infrared analysis are required for analysis of carbon monoxide[65]. In addition, colorimetric analysis is used for nitrogen oxides[66]. Mass spectrometry, gas chromatography, infrared spectroscopy, and non-dispersive infrared analysis are used for hydrocarbons[65,67]. Regarding these types of problem, we demonstrate that our artificial olfactory system based on the eigengraph waveform analysis method can be a promising alternative that can simultaneously and efficiently analyze the complex exhaust gases according to the automobile engine types and their individual gas components with similar chemical structures[68].

The target gases of the empirical experiment consist of two exhaust gases from gasoline and diesel engines, two carbon oxides (CO, $CO_2$) and two nitrogen oxides (NO, $NO_2$). The two exhaust gases depending on engine type are classified depending on the presence or absence of NO and $O_2$ components. In detail, the gasoline exhausts contain CO, $CO_2$ and $C_3H_8$ and diesel exhausts contain an additional NO and $O_2$. The information of their detailed content was shown in Supplementary Fig. 9. As we mentioned, the NO, $NO_2$ and $CO_2$, are known as oxidizing gases and the CO, $C_3H_8$ are known as reducing gases. We measured the target gases using four ORSAs and plotted their eigengraphs in Fig. 5 and Supplementary Fig. 10. A detailed description of the measurements of the exhaust gas is covered in the Methods section. The data from the 9-channel sensor array measured for each target gas was assigned to one class, forming a total of 6 classes equal to the number of target gases. As a result of the measurement, the differences in the eigengraph waveforms for each gas type were clear enough to be distinguished through the characteristic points of each waveform with the naked eye (Fig. 5, Supplementary Fig. 10). In particular, in the case of gasoline and diesel engine exhausts, very unpredictable and unique waveforms were observed due to the mixture of various reactions from individual components. This result support with our argument about the existence of eigengraph.

For a more exhaustive analysis, Euclidean similarity analysis was performed between all measurement data within the intra-class to verify the reliability and reproducibility of signal generated by different sensors (Supplementary Table 7). Euclidean distance can be calculated simply by subtracting data plots between the comparing channels. In the case of comparison between the same channels, each channel consists of 16 signal data, so the average value calculated by mutually comparing all data plots of 16 signals was used as the Euclidean distance. In the case of comparison between the different channels, the data plots of all signals between the two channels are fully compared to each other. From the Euclidean matrix, it can be seen that the main diagonal components usually have the minimum values compared to other channel values. In other words, similar reaction characteristics between gas molecules and sensing materials are stably maintained for the same channel data, so uniform signals were generated reproducibly and the Euclidean distance values were small (Supplementary Fig. 10). On the other hands, the values for different channel data were large due to their different waveforms.

In addition, to deeply classify the inter-class data and resolve concerns about overfitting in various deep learning model, fourfold cross-validation was performed on 4 sets of 4-cycle signals generated simultaneously from each of the four chips. Before the main analysis, the eigengraphs for the exhaust gases were converted into MFCC feature vectors and used as input data for deep learning networks. Basically, we performed deep learning analysis using the DNN architecture. In addition, we further verified the versatility and

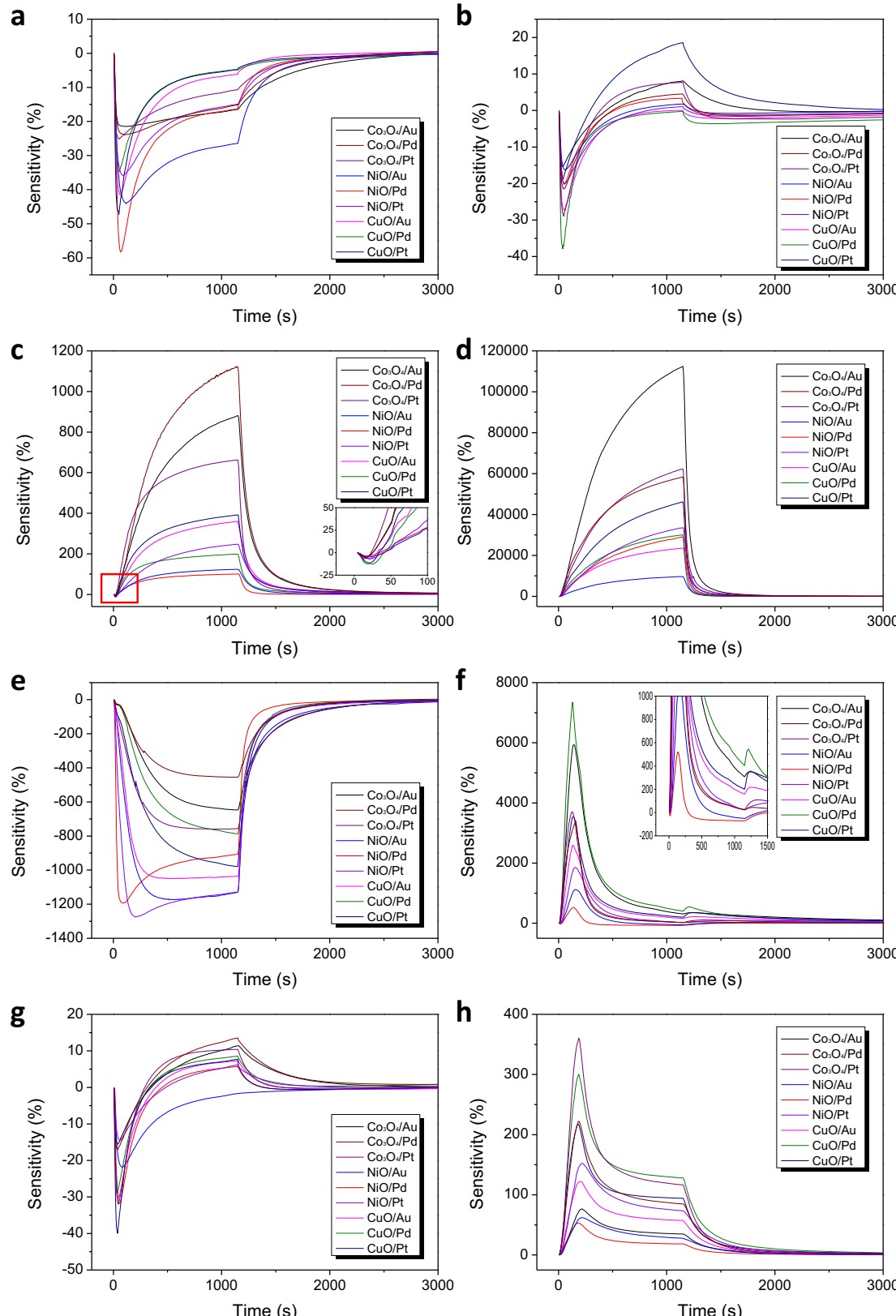

**Fig. 5 | Representative eigengraph waveforms generated through the empirical experiments on the automobile exhaust gases. a** CO. **b** CO$_2$. **c** NO. **d** NO$_2$. **e** Gasoline exhaust gas (including CO(5.01%), CO$_2$(14.0%), C$_3$CH$_8$(0.20%)). **f** Diesel exhaust gas (including NO(0.18%), CO(4.98%), CO$_2$(14.0%), C$_3$CH$_8$(0.20%) and O$_2$(1%)). **g**, CO + CO$_2$. **h** CO + CO$_2$ + NO.

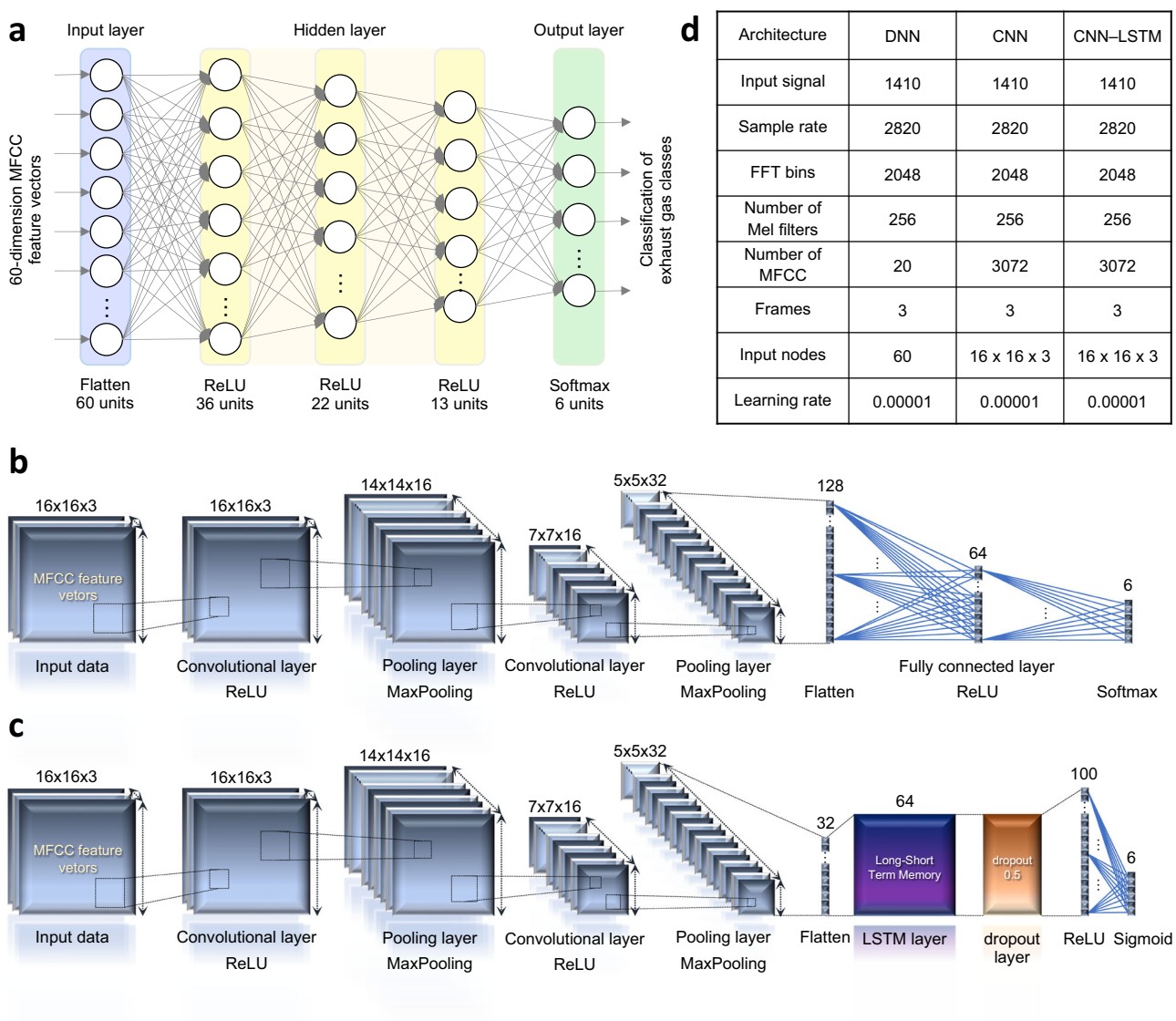

**d**

| Architecture | DNN | CNN | CNN–LSTM |
|---|---|---|---|
| Input signal | 1410 | 1410 | 1410 |
| Sample rate | 2820 | 2820 | 2820 |
| FFT bins | 2048 | 2048 | 2048 |
| Number of Mel filters | 256 | 256 | 256 |
| Number of MFCC | 20 | 3072 | 3072 |
| Frames | 3 | 3 | 3 |
| Input nodes | 60 | 16 x 16 x 3 | 16 x 16 x 3 |
| Learning rate | 0.00001 | 0.00001 | 0.00001 |

**Fig. 6 | Deep learning architectures for the empirical experiments of the automobile exhaust gases. a** DNN (Deep neural network) **b** CNN (Convolution neural network) architecture. **c** CNN-LSTM (Long-short term memory) architecture. **d** Summary of the parameter settings according to the deep learning architecture.

expandability of the MFCC feature vector through experiments using primitive deep learning networks such as convolution neural network (CNN) and long-short-term memory combined with convolution neural network (CNN-LSTM) (Fig. 6). The CNN is an architecture that extracts and learns key features for classification by preserving the spatial characteristics of input data such as time series data and image data[69]. When applied to our system based on waveform analysis, the classification performance can be improved more effectively by utilizing the characteristics of the MFCC feature vectors containing the intrinsic attributes of the eigengraph waveform. In CNN-LSTM, the CNN acts as another feature processing layers that performs additional domain transformation based on the MFCC feature vectors, and the LSTM performs training and classification on the transformed features. LSTM is an architecture that solve the vanishing gradient problem caused by long-term dependencies which was a problem when processing long sequences of time series data in recurrent neural network (RNN)[70]. As a result, the CNN feature framework with compressed sequence can be extracted while maintaining the intrinsic attributes of the MFCC feature vectors, and the converted data can be effectively classified through LSTM. The results of the fourfold cross validation using the three deep learning architectures are summarized in the

Table 1. The DNN, CNN, and CNN-LSTM showed high classification accuracy of 96.1%, 99.8%, and 100%, respectively. The CNN and CNN-LSTM architectures offer significant improvements in training time and testing accuracy compared to DNN architectures. As a result, it can be seen that the overfitting did not occur for the eigengraph deep learning models using MFCC feature vectors by achieving high test accuracies up to 99.9%, as well as training accuracies up to 100% (Supplementary Fig. 11). Additionally, we confirmed the classification performance of our system for changes in the response waveform of complex gas mixtures according to differences in components using $CO(5ppm)+CO_2(14ppm)$, $CO(5ppm)+CO_2(14ppm)+NO(2ppm)$, gasoline and diesel exhaust gases. The additional eigengraph waveforms for $CO + CO_2$ and $CO + CO_2 + NO$ have been added to Fig. 5 and Supplementary Fig. 10. The fourfold cross-validation results for the four mixed gases was shown in the Table 2. The DNN, CNN, and CNN-LSTM showed high classification accuracy of 99.3%, 99.7%, and 99.9%, respectively. The ORSAs sensitively responded to various mixed gases with different components to generate eigengraphs with various waveforms, and based on this, the deep learning software could perfectly identified the mixed gases. Consequently, we confirmed the practical applicability of our data-centric artificial olfactory system to

**Table 1 | Results of the 4-fold cross validation using the DNN, CNN and CNN-LSTM architectures for 2 exhaust gases and 4 individual components**

| Fourfold cross validation for 6 gases | | Set 1 | Set 2 | Set 3 | Set 4 | Average |
|---|---|---|---|---|---|---|
| DNN | Accuracy | 96.3% | 95.8% | 95.8% | 96.3% | 96.1% |
| | Loss | 0.11 | 0.16 | 0.10 | 0.11 | 0.12 |
| | Training Time | 5 min 3 s | 2 min 46 s | 5 min 12 s | 6 min 7 s | 4 min 47 s |
| CNN | Accuracy | 100% | 99.5% | 99.5% | 100% | 99.8% |
| | Loss | 0.019 | 0.035 | 0.061 | 0.032 | 0.037 |
| | Training Time | 1.1 s | 1.1 s | 0.8 s | 1.2 s | 1.0 s |
| CNN-LSTM | Accuracy | 100% | 100% | 100% | 100% | 100% |
| | Loss | 0.0074 | 0.0085 | 0.0081 | 0.0087 | 0.0082 |
| | Training Time | 1.8 s | 3.1 s | 1.9 s | 2.0 s | 2.2 s |

**Table 2 | Results of the fourfold cross validation using the DNN, CNN and CNN-LSTM architectures for 4 mixture gases**

| 4-fold cross validation for 4 mixture gases | | Set 1 | Set 2 | Set 3 | Set 4 | Average |
|---|---|---|---|---|---|---|
| DNN | Accuracy | 99.3% | 99.3% | 99.3% | 99.3% | 99.3% |
| | Loss | 0.023 | 0.023 | 0.094 | 0.022 | 0.041 |
| | Training Time | 25.5 s | 25.8 s | 1 min 7 s | 22.8 s | 35.3 s |
| CNN | Accuracy | 99.3% | 100% | 99.3% | 100% | 99.7% |
| | Loss | 0.041 | 0.064 | 0.054 | 0.028 | 0.047 |
| | Training Time | 0.9 s | 0.8 s | 1.3 s | 0.9 s | 1.0 s |
| CNN-LSTM | Accuracy | 99.3% | 100% | 100% | 100% | 99.9% |
| | Loss | 0.016 | 0.0016 | 0.0016 | 0.0026 | 0.013 |
| | Training Time | 1.6 s | 1.6 s | 1.6 s | 1.6 s | 1.6 s |

real-life engineering models by achieving high classification performance of various deep learning analysis for the automobile exhaust gases.

Most importantly, we emphasize that international discussions and continuous efforts for the development of adequate devices and instruments are needed to explore and secure standard eigengraphs for various odors. With a long-term vision to realize the final goal of commercialization and standardization of our artificial olfactory system, (i) various types of ORSA chips that can generate diverse eigengraphs must be developed and produced by further expanding and discovering the range of sensor materials and manufacturing methods; (ii) It is also necessary to build a gas data center that can record eigengraphs for various gases under various conditions. Based on this, (iii) commercial devices equipped with our artificial olfactory system can be developed and manufactured in connection with the data center, and real-time data measured in various unknown environments can be verified.

On the software side, subsequent work to further develop pre-trained models which automatically extract bottleneck features is recommended for actualization of standardized artificial olfactory system capable of autonomously learning and evaluating the unknown data based on sufficiently accumulated optimized datasets[58,62]. In addition, our system has versatility applicable to various artificial neural network software (Tables 1, 2) and can be advanced in forms of the system-on-chip (SoC) by integrating with the various AI accelerators such as graphics processing units (GPU) capable of high-performance computation through parallel processing or neural processing units (NPU) optimized for AI algorithm. This can enhance the computational efficiency and performance of deep learning model and contribute to saving energy costs, thereby leading to the development of real-time and low-power gas detection and various application systems based on rapid signal processing.

## Discussion

We have reported artificial olfactory system inspired by human olfactory mechanisms. The olfactory receptor-like sensor array (ORSA) was designed to make each channel have independent gas sensing characteristics to generate eigengraphs by functionalizing $SnO_2$ nanorods with combinatorial bimetallic catalysts using optimized deposition process of electron beam evaporation assisted by the spatially addressable physical shadow masks. Our manufacturing method contributed to reducing material wastage and development time, which can ultimately realize the commercialization of the highly integrated standardized ORSA chips through wafer-scale semiconductor processes. MFCC was introduced to discriminately express the intrinsic attributes of the eigengraphs reflecting the unique redox reaction characteristics between gas molecules and sensing materials. And its effectiveness was clearly demonstrated by achieving over 99.5% classification accuracy and reducing training time up to 3–4 times in deep learning analysis for the 117 labeled classes despite reducing the input size of MFCC feature vectors by up to 85.4%. In empirical experiments on automobile exhaust gases, the versatility and scalability of MFCC feature vectors were verified by achieving high test accuracy in the range of 96.1–100% in fourfold cross-validation through various deep learning architectures using DNN, CNN, and CNN-LSTM. Consequently, inspiration of the human olfactory mechanisms and device optimization through advanced nanotechnology contributed to the generation of high quality data that maximizes deep learning performance. Therefore, our data-centric approach will be expected to help to elucidate the in-depth knowledge of unique electrochemical interactions between reactive molecules or compounds and eventually be utilized not only as a promising guideline for future research of artificial olfactory technology, but also in various fields such as home appliance, humanoid robots, space exploration, defense industry and forensic investigations.

## Methods

### Fabrication of sensor array

The 3 × 3 interdigitated electrodes (IDEs) pattern of the sensor array was deposited to a thickness of Au 180 nm/Ti 20 nm on the 4-inch $SiO_2$/Si wafer by using electron beam evaporation (Supplementary Fig. 6). The number of the IDEs per each channel was 24 and the distance between each electrode was 5 μm. The vertically-ordered $SnO_2$ nanorods were deposited by using glancing angle deposition technique of electron beam evaporator. After the $SnO_2$ nanorods were deposited, the glancing angle was restored to 0 ° and the substrate rotation was stopped. Then, the transition metals (Co, Ni, Cu) and the noble metals (Pt, Pd, Au) were sequentially deposited on each row and column of the 3 × 3 IDEs pattern. The deposition specifications for the $SnO_2$ nanorods and bimetallic nanolayers were shown in the Supplementary Table 1. To crystallize the $SnO_2$ nanorods and functionalize the bimetallic nanolayers, the fabricated sensor array was annealed at

target temperature of 550 °C for 2 h while increasing temperature at 5 °C/min. Then, the 4-inch wafer was diced to obtain a total of 32 olfactory receptor-like sensor array chips.

## Characterization

The cross-sectional and plane morphology of 1D-ordered SnO$_2$ nanorods decorated with transition metal oxide and noble metal nanoparticles were characterized by scanning electron microscopy (FE-SEM, TESCAN MIRA3) at an accelerating voltage of 25 kV with a 10 mm working distance. The existence of each multi-component coated on the SnO$_2$ nanorods surface was determined by X-ray diffraction (HR-XRD, SmartLab) patterns. Electron transfer according to the type of NiO-based nanocatalysts was investigated by X-ray photoelectron spectroscopy (XPS, K-alpha). The detailed morphology of the composite nanocatalysts was inspected by transmission electron microscopy (TEM, JEOL JEM-F200) at an accelerating voltage of 200 kV.

## Gas sensing measurement for 6 VOCs

The gas sensing characteristics of the ORSA were investigated using the dedicated measurement and monitoring system that can simultaneously test up to four sensor array chips (Supplementary Fig. 7). A DC bias voltage of 1 V was applied to all sensing channels through internal probe tip arrays arranged corresponding to the IDEs pattern. Mass flow controllers controlled the concentration of the target gas by mixing dry air within a total gas flow rate of 1000 sccm. Each sensor array was loaded into a test chamber with a built-in heating plate and aged for 24 h in a dry air atmosphere at 300 °C. The target gas was injected after the dry air reaction was fully saturated and the initial baseline resistance stabilized. The resistance value of each sensing channel was simultaneously measured by switch system (Keithley 7001) which sequentially opened each channel of interface circuit in the interval of 1 s. Then, sourcemeter (Keithley 2612 A) measured the transmitted electrical response of the sensor array. Resistance values as sensing signals were recorded on a computer through the connected IEEE-488 GPIB (Supplementary Fig. 8). To build a database for deep learning analysis, the long-term sensing characteristics of the indistinguishable three gas molecules (Acetone, Toluene and Xylene) through PCA were investigated under the successive concentration changes of 2, 4, 6, 8 and 10 ppm. Furthermore, in order to verify the performance of the deep learning model for the identification of subtle differences in gas mixtures according to the mixing ratio, the three gas molecules were mixed with each other at ratios of 1:1, 1:3, 3:1 and 1:1:1. The sum of concentrations of each gas molecule constituting the mixed gases was successively set to 2, 4, 6, 8, and 10 ppm for each measurement in 1 cycle (Supplementary Fig. 4). The theoretical detection limit of each channel was calculated and shown in the Supplementary Table 2[18].

## Data preparation and feature extraction

It has been confirmed that the optimized sensor array generated a variety of eigengraphs with different range of initial resistance and fluctuation depending on the combination of its sensing materials and target gases. To compare the datasets on the same basis, resistance-based data were normalized to sensitivity values expressed in ($(R_{air} / R_{gas} - 1) \times 100\%$). As the input of the deep learning network, 117 datasets for all channels of the sensor array were labeled with binary numbers according to the gas species and mixing ratio as shown in Supplementary Table 3. Each original signal consisting of a total length of 6840 points was decomposed into 20 separate response waveforms with a length of 342 points. As a result, a total of 2370 ($9 \times 13 \times 20 = 2370$) training datasets were obtained from the responses of the 9-channel sensor array to 13 gas types.

In order to avoid distortion and loss of high-quality original signals during the reconstruction of time series signals into frequency-domain spectrum and preserve the eigencomponents, we established

a correlation between highest frequency and sampling rate for fast Fourier Transform by referring to the Nyquist-Shannon sampling theorem. This theorem suggests that the sampling rate should be at least twice the highest frequency to accurately represent the original signal[71]. Thus, the sampling rate according to the training sample length was set to 100, 400 and 800 for 50, 171 and 342 points samples, respectively. The FFT spectrum was uniformly mapped based on the frequency resolution determined by the sampling rate/FFT size. The FFT size should be usually greater than the number of input data points, and determined as the nearest multiple of $2^{72,73}$. The FFT size was set to 64, 256 and 512 for 50, 171 and 342 input data points, respectively (Supplementary Table 5). In order to secure the number of sampling points that satisfy the FFT size, the section without input data values was zero-padded to improve the frequency resolution of the FFT spectrum. The FFT spectrum represents frequency bins divided into uniform bandwidth based on the frequency resolution to linearly estimate the spectral density of the transformed frequency components such as amplitudes and phases for each frequency band[61]. And the power spectrum was generated by squaring the magnitudes of the FFT results of each frequency bin[56].

The algorithm to extract the final MFCC feature vectors from the original signals was performed automatically using the librosa package[61] and summarized as calculation of Mel-spectrogram by matrix multiplication of power spectrum and Mel-filter banks [librosa.feature.melspectrogram()], logarithmic conversion of Mel-spectrogram to generate log Mel-spectrogram [librosa.power_to_db()] and acquisition of MFCC through inverse discrete Fourier transform [librosa.feature.mfcc()]. The number of Mel filters for the 50 points sample was considered within the range of 20–40 mainly used in the speech recognition field and determined to be 32 which is a multiple of $2^{74}$. Based on this, the number of Mel filters for the 171 and 342 points samples was sequentially increased by two times[72] and set to 64, 128, respectively (Supplementary Table 5). The Mel-spectrogram was generated by the inner product of Mel-filter bank and power spectrum and the log Mel-spectrogram was obtained by taking a logarithm on Mel-spectrogram to reflect the human hearing characteristics[57]. Finally, the inverse discrete Fourier transform (IDFT) was performed to extract the MFCC feature vectors. Among the coefficients generated as many as the number of Mel filters, top 20 selected from the lowest order were used as 20-dimensional MFCC feature vectors. In the whole process, the time required for fast Fourier transform and MFCC extraction of the raw signal according to the training sample size was negligibly small.

## Deep learning model

A fully connected deep neural network (DNN) model based on a multilayer perceptron architecture with three hidden layers was designed for the classification of multiclass labeled according to gas species and mixing ratios (Supplementary Table 4). The DNN model was implemented using Keras and Tensorflow packages [tf.keras.Sequential()]. The 20-dimensional MFCC input vector was flattened into a one-dimensional array in the input layer. To prevent the vanishing gradient problem in which the intrinsic attributes of the MFCC feature vectors disappear as the training was progressed, the hidden layers were activated by the ReLU function[75]. In the output layer, the Softmax function for multiclass classification was used to estimate the prediction probability about all input classes as probability values between 0 and 1[63]. Referring to the heuristic function ($r = (\#input / \#ouput)^{1/4}$) established by previous researchers, the node value of the three hidden layers was set to 32, 50, and 80, respectively (Supplementary Table 4.). To build a system network for training the model, optimizer, loss function, and metrics factors were determined by setting the learning environment. The Adam optimization algorithm based on the stochastic gradient descent was used to optimize the deep learning model[76]. It could update the network weights to implement a

computationally efficient model by minimizing the loss function according to the training result [keras.optimizers.Adam()]. The learning rate was set to 0.0001 and 0.00001 for each analysis case comparing the three training sample sizes. The loss and metrics were set as 'sparse_categorical_crossentropy' and 'accuracy' to perform training [model.compile()]. At the same time, the model was trained according to the three training sample size, and the classification performance of the model was evaluated by calculating each training accuracy and loss [model.evaluate()] (Supplementary Table 5).

## Experiments of the exhaust gases

The two exhaust gases depending on engine type (gasoline, diesel) and their four individual components (NO, $NO_2$, CO, $CO_2$) were measured using four ORSA chips simultaneously fabricated in same manufacturing process (Fig. 2 and Supplementary Fig. 7). The exhaust gases were standard gases that have been strictly evaluated and certified by the Korean Laboratory Accreditation Scheme (KOLAS) operated by the National Institute of Technology and Standards under the Ministry of Trade under Industry and Energy of the Republic of Korea. Their specific information is provided in Supplementary Fig. 9. The KOLAS which was established in accordance with basic act on national standards is an internationally recognized organization for the reliability of accredited testing and calibration institutions, which operates various standard systems across the country and industry such as production of standard material, product certification, proficiency test operation, verification and accreditation systems. The concentrations of the two exhaust gases was diluted as 1/20 by mixing dry air within a total gas flow rate of 1,000 sccm through mass flow controllers. Four individual gases were measured at 10 ppm in a dry air atmosphere at 300 °C. The target gas was injected after the dry air reaction was fully saturated and the initial baseline resistance stabilized. The resistance value of each sensing channel was simultaneously measured by switch system (Keithley 3706) which sequentially opened each channel of interface circuit in the interval of 250 ms. Keithley 3706 can provide improved resolution through faster switching performance than the existing Keithley 7001. Keithley 3706 provides improved resolution through faster switching performance than the existing Keithley 7001, allowing us to more closely capture instantaneous changes in eigengraph waveforms. Four ORSA chips were used simultaneously to measure the target gas four times in succession at the same concentration, resulting in a total of 16-cycle gas sensing signals (Supplementary Fig. 10).

One cycle of the signal consists of a total of 1410 discrete data points. In general, the time series characteristics of gas reaction data can be divided into (i) the section where the reaction begins and converges to the peak (gas reaction section), (ii) the section where air is injected as recovery gas, the gas molecules bound to the sensing material are removed, and the sensor begins to recover to its original state from the peak (recovery section), (iii) the section where the downward trend is stable due to long-term recovery (complete recovery section). Therefore, considering the time series characteristics of the gas detection signal, the signal of all 1410 points is divided into 3 sections at intervals of 470 points and feature processing is performed for each section. The sampling rate was set to 2820, which is twice the total data points. The FFT size was set to 2048, which is a multiple of 2 greater than 1410. The number of Mel filters was set to 256. The size of the MFCC feature vector was set differently depending on the type of deep learning architecture. It was set to 20, 3072, and 3072 for DNN, CNN, and CNN-LSTM, respectively. The parameters required for feature engineering to determine the input size for deep learning are summarized in the Fig. 6.

Below, we describe the application of DNN, CNN, and CNN-LSTM deep learning architectures using the preprocessed data. First, Total 60-dimensional MFCC feature vectors were used as input data for the DNN architecture by combining the 20-dimensional MFCC extracted from three sections of each signal. The DNN architecture consists of an input layer, an output layer, and three hidden layers. The number of nodes for the three hidden layers was set to 36, 22 and 13 respectively, which were activated by the ReLU function. The final output layer activated by Softmax function distinguishes the six classes for each gas species. Second, the preprocessed 3072-dimensional MFCC feature vectors were converted into structured data with a size of $16 \times 16 \times 3$ that can be input into the CNN architecture. To extract new features specialized for CNN based on MFCC, two convolutional layers and two pooling layers were used alternately. In the convolution layer, weight calculation is performed through $3 \times 3$ filter projection and activated through the ReLU function. The pooling layer compresses the data size of previous layer by extracting key features through max pooling while maintaining the spatial characteristics of the input data. The extracted features were flattened into a one-dimensional array with a length of 64 in the flattening layer and fully connected with the output layer for classification. In the case of CNN-LSTM, features extracted from CNN are flattened and input to the LSTM layer for learning and classification. Detailed information of the architectures are shown in Fig. 6a–c.

## Data availability
The main data generated in this study have been deposited in the Figshare database [https://doi.org/10.6084/m9.figshare.24312670].

## Code availability
The code for eigengraph deep learning analysis generated in this study have been deposited in the Figshare database [https://doi.org/10.6084/m9.figshare.24312670].

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

## Acknowledgements

This work was financially supported by National R&D Program (2017M3A7B4041987, 2019R1A2C2090443, 2021M3H4A3A02086430) through the National Research Foundation of Korea (NRF) funded by the Ministry of Science and ICT (MSIT), Technology Innovation Program (20013621, Center for Super Critical Material Industrial Technology) funded by the Ministry of Trade, Industry and Energy (MOTIE), and the Technology Development Project for Biological Hazards Management in Indoor Air Program (ARQ202101038001) through Korea Environmental Industry & Technology Institute (KEITI) funded by the Ministry of Environment (ME). The Inter-University Semiconductor Research Center and Institute of Engineering Research at Seoul National University provided research facilities for this work.

## Author contributions

S.C.J. supported and supervised this project. S.-H.S. conceived this study and designed the artificial olfactory system. S.-H.S., J.M.S. and H.W.J. devised the material composition of the sensor array. S.-H.S. and J.M.S fabricated the sensor arrays and devised mass production process. S.-H.S. investigated the material characteristics and the gas sensing properties of the olfactory receptor-like sensor array. S.-H.S. and J.G.P. designed the mathematical feature engineering and the eigen-graph deep learning analysis process. S.-H.S. built the preprocessed database. J.G.P. extracted MFCC feature vectors and devised the deep neural network model to conduct deep learning analysis. S.-H.S. wrote the paper. All authors reviewed the manuscript and provided the feedback. S.C.J. supported and supervised overall revision process. S.-H.S. and J.G.P. devised the revision plans and experiments. J.M.S. and H.W.J. discussed about the materials for the gas sensor. S.-H. S. manufactured the sensor arrays and performed the empirical experiments on the exhaust gases. Y.J.H. assisted to prepare the experiments. S.-H.S. pre-processed the exhaust gas data and analyzed Euclidean similarity. J.G.P. designed the DNN, CNN and CNN-LSTM architectures. J.G.P. and S.-H.S. performed the 4-fold cross validation test through deep learning analysis. S.-H.S wrote the peer review file corresponding to the reviewer's comments. S.-H.S revised the manuscript. All authors reviewed the revised manuscript and provided the feedback.

## Competing interests

The authors declare no competing interests.
