## [Peer Review File · Nature Communications]

REVIEWER COMMENTS

Reviewer #1 (Remarks to the Author):

This manuscript seems to be well organised.

The reviewer has a strong interest in research on multimodal sensing, such as artificial skin, besides artificial olfaction. Therefore, this manuscript was carefully reviewed. It proposes a more accurate sensing system by introducing recently emerging ICT technologies.

It is sadly difficult to find originality and high scientific value in this manuscript.

As the author states in the introduction, research on sensor arrays based on electrochemical sensors started in 1982.

In the 2010s, various applied research results have been published, including Sensight's cyranose320, a monitoring sensor respiratory diagnostic device using oxide-based electrochemical sensors.[1-3]

It is difficult to find any novelty in this manuscript's metal oxide-based sensor device.

In order for this manuscript to be published in Nature communication, it is necessary to show scientific novelty or an engineering application model based on a high level of completeness, such as excellent detection limits (ultra-sensitive sensing in real life fields like diagnostic devices).

It seems that it is necessary to show the engineering value by presenting a complete application model rather than simply mapping data for the reference gas.

If this author wants to emphasise the improvement of accuracy to 99.5% by introducing an artificial intelligence learning model, a more systematic manuscript on the design of artificial intelligence analysis algorithms is needed. However, in 2023, a manuscript on the design of a neural network algorithm is still not enough to be published in Nature Communications, a top-tier journal.

For single composition gas discrimination, above the 99% accuracy is expected to be high enough to use the artificial intelligence learning tools currently provided by Origin or Matlab.

This reviewer believes that this manuscript can be published in top academic journals such as Nature Communications if it is rewritten using this group's high-level fabrication technology prepared in this manuscript and adding advanced application model research, such as respiratory diagnosis and high-sensitivity sensing.

[1]Panigrahi, S., et al. "Neural-network-integrated electronic nose system for identification of spoiled beef." LWT-Food Science and Technology 39.2 (2006): 135-145.

[2]Dragonieri, Silvano, et al. "Electronic nose technology in respiratory diseases." *Lung* 195 (2017): 157-165.

[3]Wilson, Alphus D. "Recent progress in the design and clinical development of electronic-nose technologies." *Nanobiosensors in Disease Diagnosis* 5 (2016): 15.

Reviewer #2 (Remarks to the Author):

The paper reports on the realization of an artificial olfactory system, combining semiconductor fabrication based gas sensors combined with MFCC based deep learning analysis for gas classification.

The application of MFCC based classification, usual in audio/speech processing, to 'odour' sensing seems to be the key strength of this paper. What detracts from it is the small sample size, but it is still interesting as an initial proof of concept.

While the authors claim the standard semiconductor processing of the sensing devices as a point of strength, the novelty here is more difficult to see. However, the integration of the sensing hardware with the measurement and classification certainly lends strength to this work.

There are a couple of moderate concerns I would request the authors to address.

1. The biggest concern is the claim made or sense conveyed, across several points in the manuscript (lines 102, 114, 138, 393; captions of figures saying 'olfactory receptor like'), of mimicking human olfaction. In my opinion, this work certainly qualifies as a commendable initial demonstration of an artificial olfactory system. However, it seems to only as close to human olfaction as a CMOS image sensor is to human vision - namely, they carry out the same sensory function. In my view, to convey the sense of mimicking human olfaction is a little misleading, and ought to be removed. In my mind, this would not detract from the substance of the advance reported here.

2. The authors have not formally defined and introduced the 'eigengraph'. In other words, there seems to be a gap between where Fig. 3 ends, and where Fig. 4a (sensitivity time series, detailed in Supp. Fig. 4) begins. Fig. 4 purports to illustrate the 'Eigengraph Deep Learning Process' - but the reader is not told what exactly the eigengraph is. Given the expanse of this work - which is commendable, and the wide readership of this journal, it might be useful to devote a few lines to to introduce it. This could be where it first appears in the body (line 71) or even in the Supplementary Materials section.

A few more minor comments are as follows:

3. The sentence construction in line 24 of the abstract seems to leave it a little unclear.

4. It may be a good idea to expand 'MFCC' in the abstract (line 26). On a related note, it may be useful to many readers to have a few lines of introduction to it - either around line 127, or in the Supplementary Materials section.

5. Around line 182, the authors claim that their design and manufacturing strategy can help discover optimal combination of sensing materials. In my mind, it is not clear which part of this work covers that exactly. The authors could consider adding some text to clarify. If the authors are suggesting that this is something that could be developed as a corollary or followup of this work, they could clarify that.

Reviewer #3 (Remarks to the Author):

1/ Summary of work.

In humans, the olfactory system "senses" a gas, sends message to the central nervous system which "processes" the information and identifies the gas. This work proposes to solve the gas-sensing problem with the same framework. Here, the olfactory system is replaced by a sensor array, which turns gases into some waveform signals. The central nervous system is replaced by a machine learning model which has two steps, described in lines 299 to 319:

(i) data processing. Preprocess the signals with Fourier transform, take log to get the MFCC frequency and keep the first 20 terms.

(ii) learn the mapping from (MFCC feature vectors) to (classification of 117 gases). This is a standard classification problem, and this paper uses a neural network with fixed architecture to solve it.

The authors showed that they achieved between 85% to 99.9% training accuracy depending on the size of the training set and their training time.

2/ Will the work be of significance to the field and related fields?

I am not well-versed enough in this literature to answer this question. Per the literature cited in the paper, this approach to build an artificial olfactory system seems new.

3/ Concerns and Flaws.

It seems that existing work have not considered this because reproducibility is a major concern, as, quoting the paper (lines 55 to 61):

" Even though particular sensing material has high reactivity to a specific gas molecule, the sensitivity can vary depending on the degree of diffusion of the gas molecule reaching the gas sensor. In addition, gaseous analytes often exist as complex mixtures of individual gas molecules with varying concentrations and58

chemical formulas. In this case, since the overall sensitivity can be distorted due to the amplification or attenuation of the reactivity by the competition between individual gas molecules in the mixed gas, it

is difficult to expect accurate gas identification results with the sensitivity-oriented data analysis methods"

The new method, in my understanding, claims to get around this road block as follows.

1. Have a standardized sensor array, not one specific one per compound
2. Use machine learning to identify the gases based on their different waveforms.

However, there are two reproducibility issues here:

1. Do different arrays reliably produce the same waveform for the same gas.
2. How well does the trained neural network perform out-of-sample on datasets produced by the same sensor array? How about on datasets produced by a different sensor array? What about datasets with different mixture ratio of the different gases?

Machine learning systems are notorious for over-fitting, so this is very much a concern. I do not see an extensive discussion that address these points. The paper uses the word "reproducible" several times but I do not see data that convinces the reader on this.

**Responses to Reviewers' Comments**

*Nature Communications*

October 15, 2023

Dear Reviewers,

We would like to sincerely appreciate the reviewers for the careful consideration and thorough
examination of our manuscript entitled “Data-Centric Artificial Olfactory System based on the
Eigengraph” (manuscript number NCOMMS-23-09667-T). We gained a better understanding of the
critical issues in this paper through the detailed and accurate reviewers’ comments. According to their
many helpful suggestions, we have revised our manuscript through the complementary experiments
and new analyses. We acknowledge that the scientific and scholarly quality of our manuscript was
improved by the scrutinizing efforts of the reviewers. Below we provide point-by-point responses to
the reviewer comments. The changes within the revised manuscript were highlighted (underlined
and in blue). Important descriptions of the manuscript emphasized by the author were marked
(in orange). The response has become somewhat longer due to the detailed description of the
reviewers' comments, but we would appreciate your generous understanding for the advancement of
this research and scholarship. We hope sincerely that our responses sufficiently address the reviewers’
concerns. We look forward to hearing from you in due time regarding our submission and to respond
to any further questions and comments you may have.

Sincerely,

Seung-Hyun Sung

Prof. Seong Chan Jun

School of Mechanical Engineering, Yonsei University

50 Yonsei-ro, Seodaemun-gu, Seoul 03722, Republic of Korea

Tel: +82-2-2123-5817 / e-mail: scj@yonsei.ac.kr

**REVIEWER COMMENTS**

**Reviewer #1 (Remarks to the Author):**

This manuscript seems to be well organised.

The reviewer has a strong interest in research on multimodal sensing, such as artificial skin, besides
artificial olfaction. Therefore, this manuscript was carefully reviewed. It proposes a more accurate
sensing system by introducing recently emerging ICT technologies.

It is sadly difficult to find originality and high scientific value in this manuscript.

As the author states in the introduction, research on sensor arrays based on electrochemical sensors
started in 1982.

In the 2010s, various applied research results have been published, including Sensight's cyranose320,
a monitoring sensor respiratory diagnostic device using oxide-based electrochemical sensors.[1-3]

It is difficult to find any novelty in this manuscript's metal oxide-based sensor device.

In order for this manuscript to be published in Nature communication, it is necessary to show
scientific novelty or an engineering application model based on a high level of completeness, such as
excellent detection limits (ultra-sensitive sensing in real life fields like diagnostic devices).

It seems that it is necessary to show the engineering value by presenting a complete application model
rather than simply mapping data for the reference gas.

If this author wants to emphasise the improvement of accuracy to 99.5% by introducing an artificial
intelligence learning model, a more systematic manuscript on the design of artificial intelligence
analysis algorithms is needed. However, in 2023, a manuscript on the design of a neural network
algorithm is still not enough to be published in Nature Communications, a top-tier journal.

For single composition gas discrimination, above the 99% accuracy is expected to be high enough to
use the artificial intelligence learning tools currently provided by Origin or Matlab.

This reviewer believes that this manuscript can be published in top academic journals such as Nature
Communications if it is rewritten using this group's high-level fabrication technology prepared in this
manuscript and adding advanced application model research, such as respiratory diagnosis and high-
sensitivity sensing.

[1]Panigrahi, S., et al. "Neural-network-integrated electronic nose system for identification of spoiled
beef." LWT-Food Science and Technology 39.2 (2006): 135-145.

[2]Dragonieri, Silvano, et al. "Electronic nose technology in respiratory diseases." Lung 195 (2017):
157-165.

[3]Wilson, Alphus D. "Recent progress in the design and clinical development of electronic-nose
technologies." Nanobiosensors in Disease Diagnosis 5 (2016): 15.

**1) Reviewer's comment:**

It is sadly difficult to find originality and high scientific value in this manuscript.

As the author states in the introduction, research on sensor arrays based on electrochemical sensors
started in 1982.

In the 2010s, various applied research results have been published, including Sensight's cyranose320,
a monitoring sensor respiratory diagnostic device using oxide-based electrochemical sensors.[1-3]

It is difficult to find any novelty in this manuscript's metal oxide-based sensor device.

[1]Panigrahi, S., et al. "Neural-network-integrated electronic nose system for identification of spoiled
beef." LWT-Food Science and Technology 39.2 (2006): 135-145.

[2]Dragonieri, Silvano, et al. "Electronic nose technology in respiratory diseases." Lung 195 (2017):
157-165.

[3]Wilson, Alphus D. "Recent progress in the design and clinical development of electronic-nose
technologies." Nanobiosensors in Disease Diagnosis 5 (2016): 15.

**Author's response:**

We appreciate the reviewer's comments. We acknowledge that our description for this study
may have not been sufficient to engage scientific interest of our readers. Nevertheless, we
believe that there is a lot of novelty and originality in our research that has not yet been reported
or highlighted. In response to this question, we have prepared three sections sequentially as
follows: (1) Summary of novelty and originality of the manuscript. (2) Author's comments on
the presented papers. (3) Vision and future plans for the results of this study. We hope that our
responses sufficiently address the reviewer's concerns and questions.

**(1) Summary of novelty and originality of our study.**

First, inspired by the breakthrough discovery about human olfactory mechanism acknowledged
for contribution to winning the 2004 Nobel Prize in Physiology or Medicine (by Richard Axel,
Linda B. buck), we designed our artificial olfactory system in an attempt to essentially mimic
the working mechanism of olfactory receptors and olfactory center. Based on concept of the
olfactory mechanism that olfactory receptors generate unique electrical signals from the
chemical reaction with odors, we assumed that different types of sensing materials would have
unique response properties for specific gas molecules rather than simply using multiple sensor
arrays as sensing units. Thus, breaking away from the conventional research flow of sensor
development focused on the magnitude of sensitivity, our research began with a goal to
discover new gas sensing characteristics to profile odors which reflect the waveform attributes
of the gas detection signals according to the type of redox reaction between gas molecules and
the sensing materials. As a part of that, based on the well-established theories for metal oxide-
based gas sensors and surface functionalization engineering using nanocatalysts, we made an
effort to implement novel gas sensors (called olfactory receptor-like sensor arrays (ORSAs))
with new gas detection characteristics by simultaneously combining noble metal nanocatalyst
decoration and metal oxide nano-heterojunction. To realize this, we adopted electron beam
evaporation as the fabrication method for the sensor array to take advantage of the various
manufacturing strengths of physical vapor deposition technology. Since, our devised sensor
array manufacturing process consists of an automated process with minimal human error in a
high vacuum atmosphere isolated from the outside, it ensures uniform thin film quality and
reproducible sensing performance and facilitates commercialization and standardization of our
system through mass production of wafer-level fabrication. This study was conducted by
improving our previous research case¹. The wet process such as PS bead monolayer arrays
transfer, which had caused problems with uniformity of sensing film of nanodome structure-
based sensor arrays in previous studies, was eliminated and we implemented a complete dry
process at wafer scale. Differentiation of gas sensing characteristics based on nano-structural
differences and the presence or absence of Au catalyst did not effectively contribute to
diversifying selectivity. In this study, we achieved improvement in selectivity through various
catalyst combination strategies. In this process, we elucidated the influence and role of novel
composite nanocatalysts composed of noble metals and transition metal oxides in modifying
the gas sensing characteristics.

Reference

- 1. Lee, J. et al. High-performance gas sensor array for indoor air quality monitoring: the role
of Au nanoparticles on WO₃, SnO₂, and NiO-based gas sensors. *J. Mater. Chem. A*, **9**,
1159–1167 (2021).

Second, in the process of developing and optimizing sensor arrays, we experimentally
discover that various unique signals with different waveforms exist depending on the type of
the redox reaction between gas molecules and sensing materials. Accordingly, we newly
defined major four conditions that time series gas detection signals should have, which many
researchers have overlooked for a long time. We formally call this Eigengraph and incorporate
the detailed description about eigengraph in the manuscript. (Page 11, Line 289)

**“The law of existence of eigengraph and its implementation conditions**

In this section, formal definition of the eigengraph and the conditions for generating
eigengraphs are described based on the phenomenological observation of our experimental
result of gas reaction using metal oxide-based gas sensors. Since the gas response signal of gas
sensor is the result of the interaction between the sensing material and the gas molecules, the
justification for the existence of the eigengraph is explained by considering all the theoretical
factors for each.

First, in terms of gas sensors, the receptor function and the transducer function are the key
factors explaining the principle of the gas detection mechanism and the generation of the
detection signal (as a result of the reaction)⁴⁵⁻⁴⁷. Receptor function is related to the
chemisorption, desorption and redox reaction of gas molecules on the surface of the sensing
material. The transducer function converts the results of chemical interaction with gas
molecules into electrical signals which flow through the conduction channels with the
Schottky-barriers formed by interconnection of the grain boundaries of adjacent nanoparticles.
The surface charge density of sensing materials is changed by the redox reaction that occurs
during the chemisorption of gas molecules, resulting in the formation of a charge depletion
region. This leads to the change of the height of the Schottky barrier, which changes the
resistance of the conduction channel^{47, 48}. In other words, the receptor function and the
transducer function are closely related to each other, and these surface reaction properties
basically depend on the intrinsic physical and chemical properties of the elements constituting

the sensing material, and can also be influenced by manufacturing and synthesis methods. For
example, electrical properties of semiconductor materials, such as density of states, energy
band structure, and carrier concentration, are unique depending on the type of constituent
elements. In particular, when metal or metal oxide nanocatalysts having different Fermi levels
are added, electrical properties of the sensing material are modified by carrier diffusion in the
junction region^{49, 50}. In addition, the specific surface area for chemisorption of gas molecules
can be improved through the geometric/morphological modification of the nanostructure
according to the manufacturing method, and the electronic conduction can be improved by
controlling the crystallinity to induce grain size effect⁵⁰⁻⁵².

Second, in terms of gas molecules, the theoretically established power law for the response
of metal oxide semiconductor-based gas sensors can be summarized as a function of gas partial
pressure and sensor resistance, by combining the acceptor function and the transducer
function^{53, 54}. It means that the power law can be influenced not only by the type of gas
molecules but also by the above-mentioned material factors affecting the surface reaction
properties. Judging from this, it can be reasonably inferred that the electronegativity according
to the type of element constituting the gas molecule, the type of chemical bonding such as
covalent bond and hydrogen bond, and the bonding energy according to the molecular structure
can also affect the interaction with the sensing material. Consequently, it is obvious that the
eigengraph is generated as a result of interaction between a specific sensing material and gas
molecules.

However, since the existing theories do not consider the concept of time change, so there is
a limit to predicting the actual change in sensor resistance over time. Until now, there have
been few reports of systematic experimental results on signal formation for long-time reactions
that support widely established theoretical explanation. As a result, waveforms in the generated
graphs have not been considered as important factors. Therefore, based on the above-
mentioned major four conditions for time-series gas response waveforms, we focused on
creating eigengraphs with refined waveforms by controlling our advanced nanotechnology and
measurement environments. Eventually, we experimentally found that the eigengraphs can be
created by the reaction of specific sensing material and gas molecules using optimized sensor
arrays which have 9 different and stable gas sensing characteristics. We formally called this
Eigengraphs. In order for eigengraphs to be commonly used with engineering credibility,
researchers must create an internationally standardized and optimized device and measurement

environment. Based on this, it is necessary to find and collect eigengraphs that satisfy the major
four conditions of time-series gas response data for various gas molecules under various
environmental conditions to identify unknown odors. The law of existence of eigengraphs is
applicable to all fields dealing with various molecular reactions and it will ultimately be
possible to predict unknown target molecules through eigengraphs for unknown chemical
reactions based on advanced nanotechnology.”

**Reference (Page 29, Line 837)**

- 45. Yamazoe, N. New approaches for improving semiconductor gas sensors. *Sens. Actuators B*
*Chem.* **5**, 7–19 (1991).
- 46. Yamazoe, N. Toward innovations of gas sensor technology. *Sens. Actuators B Chem.* **108**,
2–14 (2005).
- 47. Dey, A. Semiconductor metal oxide gas sensors: A review. *Mater. Mater. Sci. Eng. B* **229**,
206–217 (2018).
- 48. Barsan, N & Weimar, U. Conduction Model of Metal Oxide Gas Sensors. *J. Electroceram.*
**7**, 143–167 (2001).
- 49. Korotcenkov, G. & Cho, B.K. Metal oxide composites in conductometric gas sensors:
Achievements and challenges. *Sens. Actuators B Chem.* **244**, 182–210 (2017).
- 50. Liu, L. et al. Heteronanostructural metal oxide-based gas microsensors. *Microsyst.*
*Nanoeng.* **8**, 85 (2022).
- 51. Korotcenkov, G. The role of morphology and crystallographic structure of metal oxides in
response of conductometric-type gas sensors. *Mater. Sci. Eng. R Repts.* **61**, 1–39 (2008).
- 52. Xu, C., Tamaki, J., Miura, N. & Yamazoe, N. Grain size effects on gas sensitivity of porous
SnO₂-based elements. *Sens. Actuators B Chem.* **3**, 147–155 (1991).
- 53. Yamazoe, N. & Shimano, K. Theory of power laws for semiconductor gas sensors. *Sens.*
*Actuators B Chem.* **128**, 566–573 (2008).
- 54. Hua, Z., Li, Y., Zeng, Y. & Wu, Y. A theoretical investigation of the power-law response of
metal oxide semiconductor gas sensors I: Schottky barrier control. *Sens. Actuators B Chem.*
**225**, 1911–1919 (2018).

Third, we developed specialized signal processing and interpretation techniques to extract
meaningful data characteristics required for gas species identification considering the diverse
waveform attributes of eigengraphs. In this study, we introduced the MFCC (Mel-Frequency
Cepstral Coefficient) feature vectors based on the Fourier transform as novel gas sensing
characteristics of eigengraphs to effectively express the reaction intrinsic reaction properties
between gas molecules and nanomaterials, for the first time. The Fourier transform is a very
powerful method that converts raw time series signals into frequency-domain spectrum by
mathematically expressing various signal waveforms with even similar amplitudes as
combinations of fundamental frequency and harmonics at their unique ratios. Due to these
strengths, the Fourier transform has been applied in various engineering fields, such as X-ray
crystallography¹, communication engineering² and vibration analysis³. From the Fourier
transformed frequency-domain spectrum, the MFCC algorithm extracts features densely from
sections that contain a lot of important information of the raw time series data, and sparsely
from sections of relatively less importance. This allows efficient data processing flow in deep
learning networks by minimizing computational load such as training and inference time. In
addition, MFCC feature vectors, which are also widely used in the acoustics and voice
recognition fields that require signal processing, operate as powerful and differentiated feature
parameters for more irregular and complex speech signals depending on individual differences
in tone, timbre, and even emotion. We believe that these parameters will be more effective
when applied to gas response signals with relatively regular periodicity. In fact, the
effectiveness and validity of MFCC has been demonstrated in various deep learning
experiments using DNN (Deep Neural Network) for gas classification by achieving
classification accuracy of up to 99.9%. Therefore, we succeeded in proving the novelty of this
study by recognizing the eigengraphs labeled into 117 classes according to the combination of
gas molecules and sensing materials as independent and unique signal waveforms with a high
analysis accuracy.

As far as we know, a gas sensor combining these two nanocatalyst functionalization methods
through physical vapor deposition has not been reported, and moreover, there have been no
attempts to identify gas molecules in terms of differences between waveforms produced by
various gas sensing characteristics. In order to distinguish differences between waveforms
according to various eigengraphs, we first introduced MFCC feature vectors specialized for
expressing the intrinsic attributes of waveforms, and confirmed their validity for gas
classification through various deep learning experiments. Thus, our system can be a powerful

alternative to overcome the distortion of selectivity due to the cross-sensitivity of metal oxide
based-gas sensors and the limitations of low selectivity due to the magnitude-based gas sensing
parameters such as sensitivity, response and recovery times, which are vulnerable to external
environmental changes. Moreover, we believe that our data-centric approach will contribute to
elucidating the in-depth knowledge of unique electrochemical interactions between reactive
molecules or compounds and eventually be utilized not only as a promising guideline for future
research of artificial olfactory technology. Consequently, our artificial olfactory system will be
used for the commercialization and standardization of various fields that require investigation
of unknown environments such as AI robots, space exploration, defense industry and forensic
investigation, which existing electronic noses could not solve.

**Reference**

- 1. Cochran, W. The Fourier Method of Crystal Structure Analysis *Nature* **161**, 765 (1948).
- 2. Weinstein, S. & Ebert, P. Data Transmission by Frequency-Division Multiplexing Using
the Discrete Fourier Transform. *IEEE Trans. Commun.* **19**, 628–634 (1971).
- 3. Al-Badour, F., Sunar, M. & Cheded, L. Vibration analysis of rotating machinery using
time–frequency analysis and wavelet techniques. *Mech. Syst. Signal Process* **6**, 2083–2101
(2011).

**(2) Author's comments on the presented paper.**

We thank for bringing up the interesting and instructive papers. We read their paper carefully
and comment on some of the similarities and differences below. These papers present various
practical application and remarkable development achievement of electronic nose using
Sensight's cyranose320 based on the conductive polymer sensor arrays. Like the cyranose320,
our system also basically consists of the same device elements, such as a multiple detection
part using a sensor array, a data analysis algorithm and assistive devices and software for data
collection, transmission and storage [1-3]. But there are decisively significant difference in
terms of the usage of data features for odor identification. **Conventional electronic nose**
**technologies including cyranose320 use response patterns called 'smell print' or 'breath print'**
**which are comprised of combinations of sensitivity magnitudes generated by multi sensor array**
**to recognize odors.** These print patterns have been analyzed through statistical pattern

recognition algorithms such as PCA (principal component analysis), cross-validation, and
discriminant analysis [2], [3].

Unfortunately, the sensitivity is basically a magnitude-based parameter that is prone to
change depending on the external environment, as mentioned in the given papers as a
disadvantage of conductive polymer sensor of cyranose320 [1], [3]. Therefore, since
measurement values for target gases may be fluctuated in an unknown environment where
information on atmospheric states such as species composition, distribution, humidity, and
fluidity is obscure, it is hard to directly compare with reference values previously recorded in
a controlled environment. Eventually, it makes difficult to clearly identify and present critical
factors that affect the magnitudes of the measured parameter values.

Actually, since the conventional magnitude-based parameters such as sensitivity, response
and recovery times vulnerable to the external environmental changes will distort scale-
dependent PCA results to make the visualization difficult, the analysis results are difficult to
have high reliability in terms of reproducibility. Moreover, the PCA visualization, which is
widely used in the field of gas sensors, is nothing more than simply showing the results of the
projecting data onto a hyperplane created by an axis with the maximum variance of magnitude-
based datasets and its orthogonal axis [1]. Likewise, the variance of these datasets can vary
depending on the magnitude of data value. Thus, the PCA results do not explain any phenomena
caused by physical and chemical reactions between sensing materials and gas molecules,
making it more difficult to predict the unknown consequences. Consequently, the sensitivity
patterns work as simply indicators of the comprehensive results of the multi sensor array's
overall responses to odor, and it is difficult to infer detailed information such as about the
individual gas molecules that make up the odor from them [1]. Actually, due to these limitations
of existing electronic nose technology, the authors were concerned that problems may arise in
diagnosing diseases by detecting abnormal VOCs generated in diseased tissues, called
biomarkers that indicate specific diseases [1]. In other words, it only allows simple analysis at
the level of predicting or relating specific chemical phenomenon or a specific disease symptom
by comparing the response patterns for odors. The limitations of PCA results and the validity
of using non-parametric data analysis techniques such as artificial neural networks presented
by S. Panigrahi are consistent with our argument [1].

Exceptionally, the electronic nose has shown great achievements and potential in the medical
industry and clinical diagnostics for disease diagnosis as an affordable, portable, rapid, non-

invasive medical tool [2], [3]. The author explains the reasons why electronic nose technology
was effective in the conventional medical field as follows. Since the collection of the patient's
exhaled breath is accomplished through direct short-distance contact between the breath
analysis device and the patient or supplementary techniques for collection and storage of gas
sample such as Tedlar and Mylar bags, the influence of the external environment can be
minimized [2]. Thus, the existing analysis method using sensitivity-based breath print may be
valid.

On the other hand, our artificial olfactory system collects diverse eigengraphs for various
conditions to create database and trains deep learning models for classification of the
differences in the eigengraph waveforms using MFCC feature vectors, identifies the unknown
odors by comparing them with the reference data recorded in the database. That is, our system
can overcome many limitations of existing electronic nose technology such as confusion of
analysis results due to cross-sensitivity of individual sensor and fluctuation in sensitivity due
to unknown environmental factors by analyzing the novel gas sensing characteristics
specialized for representing intrinsic attributes of eigengraph waveforms themselves resulting
from the reaction between the sensing materials and the gas molecules. Moreover, it is possible
to perfectly identify gas molecules with similar chemical compositions and mixed gases with
various mixing ratios through the waveform analysis. The most important thing in our system
is the extraction and utilization of MFCC feature vectors. To achieve this, sensor performance
of uniform quality must be guaranteed. By producing high-quality sensing devices through the
application of semiconductor processes, it can ensure reproducibility and uniformity of data
measurement and data analysis results, and ultimately contribute to the establishment of
standard data collection techniques and standardization of data analysis techniques. Through
this, it is possible to solve the problem of standardization of breath sample collection and
analysis methods that has hindered the commercialization of the existing electronic nose
technology for respiratory diagnosis [2].

Similarly, breaking away from the existing parameters of sensitivity, response and recovery
336 times, S. Panigrahi et al. also introduced the areas under the curve calculated based on sensor
responses as new input variable features for artificial neural network (ANN) to determine beef
quality according to spoilage [1]. Although they reported improvement of the classification
performance of ANN by using the new features, the parameters still magnitude-based
parameters that does not reflect the physical and chemical reaction characteristics of

eigengraph waveforms. Besides, it has been reported that the conductive polymer used in
cyranose320 which uses sensitivity-based smell print has the disadvantage of being vulnerable
to humidity [1], [2]. Therefore, if the sensor responses decrease due to humidity, the areas under
the curve will also decrease proportionally, which may distort the detection characteristics for
the target analytes and impair analysis accuracy. **Consequently, the context in which the new
function was introduced is similar to our argument, but limitations of the area features can be
inferred in terms of stability and reproducibility against external environmental factors. In other
words, the MFCC feature vectors presented in this study have superiority in that respect as
robust and meaningful parameters with a more advanced feature engineering technology based
on the principle of waveform analysis.**

**Reference**

- [1]Panigrahi, S., et al. "Neural-network-integrated electronic nose system for identification of spoiled
beef." LWT-Food Science and Technology 39.2 (2006): 135-145.
- [2]Dragonieri, Silvano, et al. "Electronic nose technology in respiratory diseases." Lung 195 (2017):
157-165.
- [3]Wilson, Alphus D. "Recent progress in the design and clinical development of electronic-nose
technologies." Nanobiosensors in Disease Diagnosis 5 (2016): 15.

**(3) Vision and future plans for the results of this study.**

We present the objective current stage of our research and explain long-term vision to realize
our final goal including commercialization and standardization of our artificial olfactory
system. Also, relevant description has been added to the manuscript (Page 21, Line 556).

- 1. We would like to inform you that our current research is at the stage of proving the
existence of eigengraphs and presenting a new analysis method based on data-centric
approach using eigengraphs to overcome the limitations of existing methods.
- 2. We plan to fabricate various type of ORSAs by expanding and discovering the range
of sensor materials and manufacturing methods capable of generating eigengraphs and
to build a gas data center to record the eigengraphs on various gases under various
conditions.
- 3. Based on this, we will develop and manufacture commercial devices equipped with our

artificial olfactory system in connection with the data center and verify real-time data
measured in various unknown environments.

4. Ultimately, the final goal is to demonstrate our system so that it can be applied to
various environments and industries such as AI robots, space exploration, defense
industry and forensic analysis.

“With a long-term vision to realize the final goal of commercialization and standardization of
our artificial olfactory system, (i) various types of ORSA chips that can generate diverse
eigengraphs must be developed and produced by further expanding and discovering the range
of sensor materials and manufacturing methods; (ii) It is also necessary to build a gas data
center that can record eigengraphs for various gases under various conditions. Based on this,
(iii) commercial devices equipped with our artificial olfactory system can be developed and
manufactured in connection with the data center, and real-time data measured in various
unknown environments can be verified.”

**2) Reviewer’s Comment:**

In order for this manuscript to be published in Nature communication, it is necessary to show scientific
novelty or an engineering application model based on a high level of completeness, such as excellent
detection limits (ultra-sensitive sensing in real life fields like diagnostic devices).
It seems that it is necessary to show the engineering value by presenting a complete application model
rather than simply mapping data for the reference gas.

**Author’s response:**

We appreciate the reviewer’s comments. We empathize with the necessity to present scientific
novelty and an engineering application model based on a high level of completeness, such as
excellent detection limits. First, with regard to the completeness of the metal oxide gas sensor,
we have shown that the basic sensitivity and detection limit of the sensor have been greatly
improved by functionalizing the SnO₂ nanorods sensors with bimetallic nanocatalyst
combining the noble metals and transition metals in the Fig.3d (Page 3, Line 283) and

Supplementary Table 2 (Page 16, Line260). Also, the related explanations can be found in the
 Results section entitled “Investigation of the preliminary gas selectivity using principal
 component analysis” (Page 9, Line 260).

**Figure 6d**

**Figure 6e**

**Supplementary Table 2. Theoretical detection limits of all channels of the**
 **olfactory receptor-like sensor array for the indistinguishable three gas**
 **molecules through principal component analysis.**

Channel		CH1	CH2	CH3	CH4	CH5	CH6	CH7	CH8	CH9
Acetone	rms	0.6899	0.05936	0.1194	0.2809	0.07723	0.01967	0.06851	0.03011	0.007618
	slope	249.4280	54.08405	104.08854	61.2680	38.3320	22.1546	24.9290	9.9935	5.3224
	DL (ppb)	8.2982	3.2928	3.4433	13.7556	6.04483	2.6642	8.2470	9.03792	4.2904
Toluene	rms	0.3833	0.01597	0.007352	0.07738	0.02759	0.01307	0.04113	0.01453	0.01341
	slope	249.9857	46.3132	92.05123	61.7451	37.2128	21.1644	47.01643	22.9712	8.5229
	DL (ppb)	4.6001	1.03489	0.2396	3.7597	2.2249	1.8531	2.6246	1.8981	4.7214
Xylene	rms	0.3468	0.03765	0.1165	0.1677	0.1031	0.03248	0.06930	0.01793	0.01491
	slope	456.5366	77.4479	139.4088	95.8966	57.57	30.6187	86.3346	36.5372	8.0903
	DL (ppb)	2.2794	1.4585	2.5084	5.2482	5.3771	3.1832	2.4081	1.4722	5.5321

Nevertheless, we also present the limitations of poor performance of PCA due to similar
 response patterns caused by cross-sensitivity (Fig. 3e (Page 10, Line 283)). Conventional
 factors such as sensitivity and detection limit are obviously important factors in the
 completeness of the sensor, but they can easily fluctuate due to changes in the measurement
 environment, cross-sensitivity and the values themselves do not closely describe the reaction
 characteristics between sensing materials and gas molecules. **Therefore, the key factor that we**
 **emphasize through this study is the development of an olfactory system based on the analysis**
 **principle for various waveforms generated from gas reaction characteristics.** Through this
 strategy, we claim that we can solve existing chronic problems that arise from using size-based
 features such as sensitivity, response and recovery times. This strategy is based on the scientific
 discovery that eigengraphs exist between the sensing materials and gas molecules, and in order
 to create such the eigengraphs, samples with uniform quality should be produced through a
 reproducible, fully automated process with minimal human error.

To address this, we performed empirical experiments on exhaust gases to demonstrate the

scientific findings and reliability of our aforementioned claims and their applicability to the
engineering model of our proposed system. To represent the potential engineering application
and values, we have added a new section entitled “Empirical experiments on automobile
exhaust gases” (Page 16, Line 446). **Through this experiment, we were able to fully prove the**
**“The law of existence of Eigengraph” that we claimed based on scientific and engineering**
**theories in the previous section.** From the experiment that started with the reviewer's helpful
suggestion, we were able to observe various eigengraphs with various waveforms due to the
complex components contained in the exhaust gas (Fig. 5 (Page 17, Line 476) and
Supplementary Fig. 10 (Page 13, Line213)).

[revised manuscript text omitted]

From the Fig. 5 (Page 17, Line 476) and Supplementary Fig. 10 (Page 13, Line213), we
 demonstrated that various eigengraphs can be generated through various reaction
 characteristics depending on the type of gas molecules and reactants. In particular, the mixed
 gases showed unique waveforms graph that seemed to be a mixture of various reactions
 depending on their individual components. Also, we could check uniformity and
 reproducibility of the data generated from different samples fabricated by our excellent
 manufacturing process. In addition, the applicability and potential of our system based on the
 MFCC feature vectors which are specialized for characterization of various eigengraph
 waveforms toward engineering models for gas analysis were confirmed through various deep
 learning analysis on the exhaust gases using DNN, CNN and CNN-LSTM (Fig5. (Page 17,
 Line 481). The results of deep learning analysis of two types of exhaust gases according to
 engine type such as diesel and gasoline and their individual components were presented in
 Table 1 (Page 20, Line 549). Our system classified exhaust gas types and their individual
 components with similar chemical structures with up to 100% accuracy.

Moreover, the results of deep learning analysis of two types of exhaust gases according to
 engine type and mixed gases of the four individual components (CO+CO₂, CO+CO₂+NO) were
 presented in Table 2 (Page 20, Line 551). Likewise, our system showed classification accuracy
 in the range of 99.3% to 99.9% for four gas mixtures. It can be seen that the MFCC feature
 vector introduced as a new gas detection characteristic effectively contributed to gas
 identification by differentially expressing various gas response characteristics.

**Table 1. Results of the 4-fold cross validation using the DNN, CNN and CNN-**
 **LSTM architectures for 2 exhaust gases and 4 individual components.**

4-fold cross validation for 6 gases		Set 1	Set 2	Set 3	Set 4	Average
DNN	Accuracy	96.3 %	95.8 %	95.8 %	96.3 %	96.1 %
	Loss	0.11	0.16	0.10	0.11	0.12
	Training Time	5 min 3 sec	2 min 46 sec	5 min 12 sec	6 min 7 sec	4 min 47 sec
CNN	Accuracy	100 %	99.5 %	99.5 %	100 %	99.8%
	Loss	0.019	0.035	0.061	0.032	0.037
	Training Time	1.1 sec	1.1 sec	0.8 sec	1.2 sec	1.0 sec
CNN-LSTM	Accuracy	100 %	100 %	100 %	100 %	100%
	Loss	0.0074	0.0085	0.0081	0.0087	0.0082
	Training Time	1.8 sec	3.1 sec	1.9 sec	2.0 sec	2.2 sec

590 **Table 2. Results of the 4-fold cross validation using the DNN, CNN and CNN-**
 591 **LSTM architectures for 4 mixture gases.**

4-fold cross validation for 4 mixture gases		Set 1	Set 2	Set 3	Set 4	Average
DNN	Accuracy	99.3 %	99.3 %	99.3 %	99.3 %	99.3 %
	Loss	0.023	0.023	0.094	0.022	0.041
	Training Time	25.5 sec	25.8 sec	1 min 7 sec	22.8 sec	35.3 sec
CNN	Accuracy	99.3 %	100 %	99.3 %	100 %	99.7 %
	Loss	0.041	0.064	0.054	0.028	0.047
	Training Time	0.9 sec	0.8 sec	1.3 sec	0.9 sec	1.0 sec
CNN-LSTM	Accuracy	99.3 %	100 %	100 %	100 %	99.9 %
	Loss	0.016	0.0016	0.0016	0.0026	0.013
	Training Time	1.6 sec	1.6 sec	1.6 sec	1.6 sec	1.6 sec

**Figure 6**

**Fig. 2 | Deep learning architectures for the empirical experiments of the automobile exhaust gases. a.**

**DNN (Deep neural network) b. CNN (Convolution neural network) architecture. c. CNN-LSTM (Long-short term**

**memory) architecture. d. Summary of the parameter settings according to the deep learning architecture.**

The engineering value that can be obtained from the results is described as follows.
Previously, in order to accurately determine individual components of exhaust gas, different
professional analysis methods were required for each component, which was a problem that
took a lot of time and cost. For these problems, our artificial olfactory system based on the
principle of intrinsic graph waveform analysis will be a promising alternative that can
simultaneously and efficiently analyze individual gas species and mixed gases with complex
mixing ratios at once. In addition, our system can contribute to identifying fatal pollution
sources according to engine type in urban areas where air pollution due to exhaust gas is severe.
Additionally, by contributing to the enactment of appropriate laws and regulations regarding
specific exhaust gas emissions, it will prevent air pollution and improve human health and
quality of life. If our system is applied to clinical medical devices, it will contribute to
diagnosing diseases as a non-invasive tool by tracking various eigengraphs generated about the
patient's exhalation according to the disease. Since a patient's exhaled breath contains a wide
variety of gas molecular species, including biomarkers, the disease will be diagnosed by
collecting and analyzing complex waveforms from the exhaled breath itself. In addition, the
waveform of exhaled breath according to disease can be converted into a database and applied
to a wide range of medical diagnosis fields.

Therefore, we explain that the optimized entire processes required to perform data mapping
(such as sensor array fabrication and optimization for high-quality data production, MFCC
feature vector extraction based on mathematical algorithms, etc.) is a very important and
meaningful task for building a standard database to realize the Data-Central AI that the author
emphasizes. From a long-term perspective, the establishment of a standard database through
the development of standard sensor arrays and the standardization and advancement of
measurement systems is a measure and starting point for the standardization and
commercialization of fundamental artificial olfactory technology.

Separately from this, large home appliance companies, including LG Electronics, paid
attention to our research. They proposed a promising research project to apply and
commercialize our artificial olfactory system technology to products such as kimchi
refrigerators. For example, by collecting and analyzing eigengraphs for fine gas species that
occur depending on the ripening degree of kimchi, it is expected that it will be possible to
characterize the ripening degree of kimchi according to the eigengraph waveform. Instead, we
will introduce the detailed research results that we will achieve by carrying out this research

project in a follow-up paper. We apologize for not being able to introduce the completed model
right now because the research project is currently in the negotiation stage.

**3) Reviewer's Comment:**

If this author wants to emphasise the improvement of accuracy to 99.5% by introducing an artificial
intelligence learning model, a more systematic manuscript on the design of artificial intelligence
analysis algorithms is needed. However, in 2023, a manuscript on the design of a neural network
algorithm is still not enough to be published in Nature Communications, a top-tier journal.

For single composition gas discrimination, above the 99% accuracy is expected to be high enough to
use the artificial intelligence learning tools currently provided by Origin or Matlab.

**Author's Comment:**

We appreciate the reviewer's shrewd and meticulous comments. We have written this response
with serious consideration of the reviewer's comments that may relate to the fundamental and
core strategy of this study. Artificial intelligence analysis is usually performed through
algorithms and data, and improving the analysis performance of artificial intelligence can be
achieved by improving the code that makes up the algorithm (Model-centric AI) or improving
data quality (Data-centric AI). However, there are limits to increasing analysis accuracy with
the former method. Actually, the deep learning has fundamental limitations because it only
performs the role of a simple classifier that is nothing more than hierarchical expression
learning that performs geometric transformations to map the input space to the target space¹.
In addition, this model-centric AI trend so far has been due to the inability to secure abundant
high-quality data, and has continued in order to effectively learn and evaluate the small
amounts of low-quality data. **Therefore, we did not consider the model-centric approach due
to these limitations. Rather, we intended to show that high performance on gas classification
can be achieved through well-known basic deep learning algorithms by using high-quality
MFCC feature vectors generated from the highly refined eigengraphs. The "data-centric" used
in this paper title was also intended to overcome the limitation of the data utilization and
applicability due to conventional magnitude-based parameters in the gas sensor research field
by using MFCC feature vectors as novel gas sensing characteristics, and to verify the versatility**

and expandability of MFCC feature vectors in various deep learning networks for identification
of the odor attributes.

The most important novelty of this study was to develop new gas detection characteristics
that can closely characterize eigengraph waveforms, breaking away from the existing research
flow focused on developing high-sensitivity sensors and existing size-based gas detection
characteristics. It is based on the scientific fact that the eigengraphs exist due to reactions
between the sensing material and the gas molecules, which has been clearly proven for the
existence of eigengraphs through empirical experiments on exhaust gases. We added the
importance of the feature processing using the Fourier transformation-based MFCC in the
manuscript (Page 5, Line 130).

Reference

1. Chollet, F. *Deep learning with python*. (Manning publication, 2017).

“The Fourier transformation is mathematically expressed as an infinite series of products of
the complex exponential functions and their weights (magnitudes of periodic function), which
can be decomposed into the complex number of the sinusoidal periodic by Euler's formula³⁵.
That is, the key value of utilizing the Fourier transform is that any time-domain signal can be
transformed into the frequency-domain components which represent as a unique linear
combination of frequency components consisting of sine and cosine functions with various
magnitude and phase. Accordingly, Fourier transform has been widely applied in various fields
requiring signal processing such as communication engineering, vibration analysis, and speech
recognition^{36, 37}. Therefore, the Fourier transform can be an attractive method to explore the
potential intrinsic attributes of gas sensing properties from eigengraphs with various
waveforms generated by the unique redox reactions between sensing materials and gas
molecules. MFCC are feature vectors developed by imitating the human audible frequency
band caused by the nonlinear structure of the cochlea to characterize the speech signal, which
can be used as a dimensionality reduction method that selects only core features representing
eigengraphs from Fourier transformed frequency components. Finally, in the deep learning
stage, the MFCC feature vectors were input into the deep neural network model which learned
them to classify the identities of mixed gas molecules.”

**Reference (Page 25, Line 826)**

35. Bracewell, R. N. *The Fourier Transform and Its Applications 3rd Edition* (McGraw-Hill,
 2000).

36. Weinstein, S. & Ebert, P. Data Transmission by Frequency-Division Multiplexing Using
 the Discrete Fourier Transform. *IEEE Trans. Commun.* **19**, 628–634 (1971).

37. Al-Badour, F., Sunar, M. & Cheded, L. Vibration analysis of rotating machinery using
 time–frequency analysis and wavelet techniques. *Mech. Syst. Signal Process* **6**, 2083–2101
 (2011).

As we mentioned above, the current stage is to verify the performance of MFCC feature
 vectors for gas identification tasks through various deep learning. As future follow-up work on
 improving the algorithm, we plan to develop special layers for extraction of bottleneck features
 to enable unsupervised learning and evaluation, in order to overcome the limitations of existing
 neural networks. Actually, the validity, versatility, and expandability of the MFCC feature
 vectors were sufficiently verified by achieving high classification accuracy up to 100% through
 exhaust gas empirical experiments using various deep learning architectures (DNN, CNN and
 CNN-LSTM) for gas species classification (Fig5. (Page 17, Line 481).

**Table 1. Results of the 4-fold cross validation using the DNN, CNN and CNN-**
 **LSTM architectures for 2 exhaust gases and 4 individual components.**

4-fold cross validation for 6 gases		Set 1	Set 2	Set 3	Set 4	Average
DNN	Accuracy	96.3 %	95.8 %	95.8 %	96.3 %	96.1 %
	Loss	0.11	0.16	0.10	0.11	0.12
	Training Time	5 min 3 sec	2 min 46 sec	5 min 12 sec	6 min 7 sec	4 min 47 sec
CNN	Accuracy	100 %	99.5 %	99.5 %	100 %	99.8%
	Loss	0.019	0.035	0.061	0.032	0.037
	Training Time	1.1 sec	1.1 sec	0.8 sec	1.2 sec	1.0 sec
CNN-LSTM	Accuracy	100 %	100 %	100 %	100 %	100%
	Loss	0.0074	0.0085	0.0081	0.0087	0.0082
	Training Time	1.8 sec	3.1 sec	1.9 sec	2.0 sec	2.2 sec

**Table 2. Results of the 4-fold cross validation using the DNN, CNN and CNN-**
 **LSTM architectures for 4 mixture gases.**

4-fold cross validation for 4 mixture gases		Set 1	Set 2	Set 3	Set 4	Average
DNN	Accuracy	99.3 %	99.3 %	99.3 %	99.3 %	99.3 %
	Loss	0.023	0.023	0.094	0.022	0.041
	Training Time	25.5 sec	25.8 sec	1 min 7 sec	22.8 sec	35.3 sec
CNN	Accuracy	99.3 %	100 %	99.3 %	100 %	99.7 %
	Loss	0.041	0.064	0.054	0.028	0.047
	Training Time	0.9 sec	0.8 sec	1.3 sec	0.9 sec	1.0 sec
CNN-LSTM	Accuracy	99.3 %	100 %	100 %	100 %	99.9 %
	Loss	0.016	0.0016	0.0016	0.0026	0.013
	Training Time	1.6 sec	1.6 sec	1.6 sec	1.6 sec	1.6 sec

**Figure 6**

**Fig. 3 | Deep learning architectures for the empirical experiments of the automobile exhaust gases. a,**

**DNN (Deep neural network) b, CNN (Convolution neural network) architecture. c, CNN-LSTM (Long-short term**

**memory) architecture. d, Summary of the parameter settings according to the deep learning architecture**

We completely agree that it is possible to achieve a single gas accuracy of over 99% with the
artificial intelligence learning tools provided by Origin and Matlab, but direct comparison is
difficult because specific artificial intelligence algorithms were not presented. The advantages
of using Python in performing deep learning are described as follows.

First, the implementation of deep learning becomes flexible when using the Python-based
Libraries. Since Python provides various libraries such as Keras, TensorFlow, PyTorch, and
scikit-learn to make programming for artificial intelligence analysis easy. Therefore, it is much
easier and simpler to accept and handle various well-established learning methodologies of
deep learning compared to MATLAB and Origin. In particular, it provides various libraries for
feature processing and feature engineering such as librosa for extracting MFCC feature vectors,
so it has the advantage of being able to flexibly use various tools to analyze features.

Second, due to these advantages of Python as an open source development tool, it has many
users and allows rapid improvement, progress, and sharing of advanced artificial intelligence
algorithms. In fact, Google Colab provides a GPU-based computation engine, providing easy
collaboration and fast computation among various users. However, when applying artificial
intelligence learning tools, Matlab and Origin are paid commercial programs, so there are
several shortcomings in terms of usability. The lack of users makes exchange of opinions and
feedback between researchers difficult, making algorithmic progress difficult. As a result, the
ease of developing artificial intelligence may decrease.

**REVIEWER COMMENTS**

**Reviewer #2 (Remarks to the Author):**

The paper reports on the realization of an artificial olfactory system, combining semiconductor
fabrication based gas sensors combined with MFCC based deep learning analysis for gas
classification.

The application of MFCC based classification, usual in audio/speech processing, to 'odour' sensing
seems to be the key strength of this paper. What detracts from it is the small sample size, but it is still
interesting as an initial proof of concept.

While the authors claim the standard semiconductor processing of the sensing devices as a point of
strength, the novelty here is more difficult to see. However, the integration of the sensing hardware
with the measurement and classification certainly lends strength to this work.

There are a couple of moderate concerns I would request the authors to address.

1. The biggest concern is the claim made or sense conveyed, across several points in the manuscript
(lines 102, 114, 138, 393; captions of figures saying 'olfactory receptor like'), of mimicking human
olfaction. In my opinion, this work certainly qualifies as a commendable initial demonstration of an
artificial olfactory system. However, it seems to only as close to human olfaction as a CMOS image
sensor is to human vision - namely, they carry out the same sensory function. In my view, to convey
the sense of mimicking human olfaction is a little misleading, and ought to be removed. In my mind,
this would not detract from the substance of the advance reported here.

2. The authors have not formally defined and introduced the 'eigengraph'. In other words, there seems
to be a gap between where Fig. 3 ends, and where Fig. 4a (sensitivity time series, detailed in Supp.
Fig. 4) begins. Fig. 4 purports to illustrate the 'Eigengraph Deep Learning Process' - but the reader is
not told what exactly the eigengraph is. Given the expanse of this work - which is commendable, and
the wide readership of this journal, it might be useful to devote a few lines to to introduce it. This
could be where it first appears in the body (line 71) or even in the Supplementary Materials section.

A few more minor comments are as follows:

3. The sentence construction in line 24 of the abstract seems to leave it a little unclear.

4. It may be a good idea to expand 'MFCC' in the abstract (line 26). On a related note, it may be useful
to many readers to have a few lines of introduction to it - either around line 127, or in the
Supplementary Materials section.

5. Around line 182, the authors claim that their design and manufacturing strategy can help discover
optimal combination of sensing materials. In my mind, it is not clear which part of this work covers
that exactly. The authors could consider adding some text to clarify. If the authors are suggesting that
this is something that could be developed as a corollary or followup of this work, they could clarify
that.

**1) Reviewer's comment:**

The biggest concern is the claim made or sense conveyed, across several points in the manuscript
(lines 102, 114, 138, 393; captions of figures saying 'olfactory receptor like'), of mimicking human
olfaction. In my opinion, this work certainly qualifies as a commendable initial demonstration of an
artificial olfactory system. However, it seems to only as close to human olfaction as a CMOS image
sensor is to human vision - namely, they carry out the same sensory function. In my view, to convey
the sense of mimicking human olfaction is a little misleading, and ought to be removed. In my mind,
this would not detract from the substance of the advance reported here.

**Authors' response:**

We appreciate the reviewer's helpful comments. We believe that the reviewer's thorough and
accurate review will improve the completeness of our paper. **The expression of mimicking
human olfaction has been alleviated to simply simulating the sensory function of the olfactory
organ.** Based on this, we have corrected the sentences in which the reviewer concerned.

[1] Lines 102

“Our artificial olfactory system has various functional similarities because it was developed
with inspiration of the olfactory mechanism (or function) of the human body. The components

of this system corresponding to each olfactory organ of the human body are summarized as
follows (Fig. 1).” (Page 4, Line 105)

[2] Lines 114

“Second, the measurement and monitoring system inspired by the role of the olfactory bulb
and olfactory tract serve as a pathway for relaying and transmitting the generated signals.”
(Page 4, Line 117)

[3] Lines 138

Since the top-down deposition technique provides thin films with uniform quality over a large
area, it is advantageous for securing the normality, reproducibility and stability of the sensor
signal and potentially designing a stable international standard detector²⁷. (Page 5, Line 155)

[4] Lines 393

Consequently, inspiration from the human olfactory mechanisms and device optimization
through advanced nanotechnology contributed to the generation of high quality data that
maximizes deep learning performance. (Page 21, Line 588)

**2) Reviewer’s comments:**

The authors have not formally defined and introduced the 'eigengraph'. In other words, there seems to
be a gap between where Fig. 3 ends, and where Fig. 4a (sensitivity time series, detailed in Supp. Fig. 4)
begins. Fig. 4 purports to illustrate the 'Eigengraph Deep Learning Process' - but the reader is not told
what exactly the eigengraph is. Given the expanse of this work - which is commendable, and the wide
readership of this journal, it might be useful to devote a few lines to to introduce it. This could be where
it first appears in the body (line 71) or even in the Supplementary Materials section.

**Authors’ response:**

We appreciate the reviewer's comments. Following helpful suggestions from reviewers, we
added a new section to formally define eigengraphs. Following the reviewer's helpful advice,
we are very pleased with the improvement in the scientific novelty and scholarly completeness
of the manuscript.

**“The law of existence of eigengraph and its implementation conditions**

In this section, formal definition of the eigengraph and the conditions for generating
eigengraphs are described based on the phenomenological observation of our experimental
result of gas reaction using metal oxide-based gas sensors. Since the gas response signal of gas
sensor is the result of the interaction between the sensing material and the gas molecules, the
justification for the existence of the eigengraph is explained by considering all the theoretical
factors for each.

First, in terms of gas sensors, the receptor function and the transducer function are the key
factors explaining the principle of the gas detection mechanism and the generation of the
detection signal (as a result of the reaction)⁴⁵⁻⁴⁷. Receptor function is related to the
chemisorption, desorption and redox reaction of gas molecules on the surface of the sensing
material. The transducer function converts the results of chemical interaction with gas
molecules into electrical signals which flow through the conduction channels with the
Schottky-barriers formed by interconnection of the grain boundaries of adjacent nanoparticles.
The surface charge density of sensing materials is changed by the redox reaction that occurs
during the chemisorption of gas molecules, resulting in the formation of a charge depletion
region. This leads to the change of the height of the Schottky barrier, which changes the
resistance of the conduction channel^{47, 48}. In other words, the receptor function and the
transducer function are closely related to each other, and these surface reaction properties
basically depend on the intrinsic physical and chemical properties of the elements constituting
the sensing material, and can also be influenced by manufacturing and synthesis methods. For
example, electrical properties of semiconductor materials, such as density of states, energy
band structure, and carrier concentration, are unique depending on the type of constituent
elements. In particular, when metal or metal oxide nanocatalysts having different Fermi levels
are added, electrical properties of the sensing material are modified by carrier diffusion in the
junction region^{49, 50}. In addition, the specific surface area for chemisorption of gas molecules
can be improved through the geometric/morphological modification of the nanostructure
according to the manufacturing method, and the electronic conduction can be improved by
controlling the crystallinity to induce grain size effect⁵⁰⁻⁵².

Second, in terms of gas molecules, the theoretically established power law for the response
of metal oxide semiconductor-based gas sensors can be summarized as a function of gas partial
pressure and sensor resistance, by combining the acceptor function and the transducer

function^{53, 54}. It means that the power law can be influenced not only by the type of gas
molecules but also by the above-mentioned material factors affecting the surface reaction
properties. Judging from this, it can be reasonably inferred that the electronegativity according
to the type of element constituting the gas molecule, the type of chemical bonding such as
covalent bond and hydrogen bond, and the bonding energy according to the molecular structure
can also affect the interaction with the sensing material. Consequently, it is obvious that the
eigengraph is generated as a result of interaction between a specific sensing material and gas
molecules.

However, since the existing theories do not consider the concept of time change, so there is
a limit to predicting the actual change in sensor resistance over time. Until now, there have
been few reports of systematic experimental results on signal formation for long-time reactions
that support widely established theoretical explanation. As a result, waveforms in the generated
graphs have not been considered as important factors. Therefore, based on the above-
mentioned major four conditions for time-series gas response waveforms, we focused on
creating eigengraphs with refined waveforms by controlling our advanced nanotechnology and
measurement environments. Eventually, we experimentally found that the eigengraphs can be
created by the reaction of specific sensing material and gas molecules using optimized sensor
arrays which have 9 different and stable gas sensing characteristics. We formally called this
Eigengraphs. In order for eigengraphs to be commonly used with engineering credibility,
researchers must create an internationally standardized and optimized device and measurement
environment. Based on this, it is necessary to find and collect eigengraphs that satisfy the major
four conditions of time-series gas response data for various gas molecules under various
environmental conditions to identify unknown odors. The law of existence of eigengraphs is
applicable to all fields dealing with various molecular reactions and it will ultimately be
possible to predict unknown target molecules through eigengraphs for unknown chemical
reactions based on advanced nanotechnology.”

**Reference (Page 29, Line 837)**

- 45. Yamazoe, N. New approaches for improving semiconductor gas sensors. *Sens. Actuators B*
*Chem.* **5**, 7–19 (1991).
- 46. Yamazoe, N. Toward innovations of gas sensor technology. *Sens. Actuators B Chem.* **108**,
2–14 (2005).

- 47. Dey, A. Semiconductor metal oxide gas sensors: A review. *Mater. Mater. Sci. Eng. B* **229**,
206–217 (2018).
- 48. Barsan, N & Weimar, U. Conduction Model of Metal Oxide Gas Sensors. *J. Electroceram.*
7, 143–167 (2001).
- 49. Korotcenkov, G. & Cho, B.K. Metal oxide composites in conductometric gas sensors:
Achievements and challenges. *Sens. Actuators B Chem.* **244**, 182–210 (2017).
- 50. Liu, L. et al. Heteronanostructural metal oxide-based gas microsensors. *Microsyst.*
Nanoeng. **8**, 85 (2022).
- 51. Korotcenkov, G. The role of morphology and crystallographic structure of metal oxides in
response of conductometric-type gas sensors. *Mater. Sci. Eng. R Reps.* **61**, 1–39 (2008).
- 52. Xu, C., Tamaki, J., Miura, N. & Yamazoe, N. Grain size effects on gas sensitivity of porous
SnO₂-based elements. *Sens. Actuators B Chem.* **3**, 147–155 (1991).
- 53. Yamazoe, N. & Shimanoe, K. Theory of power laws for semiconductor gas sensors. *Sens.*
Actuators B Chem. **128**, 566–573 (2008).
- 54. Hua, Z., Li, Y., Zeng, Y. & Wu, Y. A theoretical investigation of the power-law response of
metal oxide semiconductor gas sensors I: Schottky barrier control. *Sens. Actuators B Chem.*
225, 1911–1919 (2018).

**3) Reviewer's comments:**

The sentence construction in line 24 of the abstract seems to leave it a little unclear.

**Author' response:**

We appreciate the reviewer's comments. We modified that sentence construction clearly.

“In addition, the physicochemically optimized sensor array contributed to maintaining the
invariance of the input data space of the deep learning model for high-performance
classification by reproducibly generating eigengraphs which have highly refined waveforms.”
(Page 1, Line 24)

**4) Reviewer's comments:**

It may be a good idea to expand 'MFCC' in the abstract (line 26). On a related note, it may be useful to
many readers to have a few lines of introduction to it - either around line 127, or in the Supplementary
Materials section.

**Author' response:**

We appreciate the reviewer's comments. Following the reviewer's advice, we have expanded
the MFCC in line 26 of the abstract. Additional detailed explanation of the Fourier transform
and the MFCC feature vector have been added on the Page 5, Line 131.

[1] Lines 26

“The effectiveness of the Mel frequency cepstral coefficient (MFCC) feature vectors in deep learning
for gas classification was clearly demonstrated in terms of training time reduction and inference
performance improvement despite the extremely decreased amount of training data.” (Page 1, Line 29)

[2] Lines 127

“The Fourier transformation is mathematically expressed as an infinite series of products of
the complex exponential functions and their weights (magnitudes of periodic function), which
can be decomposed into the complex number of the sinusoidal periodic by Euler's formula³⁵.
That is, the key value of utilizing the Fourier transform is that any time-domain signal can be
transformed into the frequency-domain components which represent as a unique linear
combination of frequency components consisting of sine and cosine functions with various
magnitude and phase. Accordingly, Fourier transform has been widely applied in various fields
requiring signal processing such as communication engineering, vibration analysis, and speech
recognition^{36, 37}. Therefore, the Fourier transform can be an attractive method to explore the
potential intrinsic attributes of gas sensing properties from eigengraphs with various
waveforms generated by the unique redox reactions between sensing materials and gas
molecules. MFCC are feature vectors developed by imitating the human audible frequency
band caused by the nonlinear structure of the cochlea to characterize the speech signal, which
can be used as a dimensionality reduction method that selects only core features representing
eigengraphs from Fourier transformed frequency components. Finally, in the deep learning

stage, the MFCC feature vectors were input into the deep neural network model which learned
them to classify the identities of mixed gas molecules.”

**Reference (Page 25, Line 826)**

4. Bracewell, R. N. *The Fourier Transform and Its Applications 3rd Edition* (McGraw-Hill,
2000).

5. Weinstein, S. & Ebert, P. Data Transmission by Frequency-Division Multiplexing Using
the Discrete Fourier Transform. *IEEE Trans. Commun.* **19**, 628–634 (1971).

6. Al-Badour, F., Sunar, M. & Cheded, L. Vibration analysis of rotating machinery using
time–frequency analysis and wavelet techniques. *Mech. Syst. Signal Process* **6**, 2083–2101
(2011).

**5) Reviewer’ comments:**

Around line 182, the authors claim that their design and manufacturing strategy can help discover
optimal combination of sensing materials. In my mind, it is not clear which part of this work covers
that exactly. The authors could consider adding some text to clarify. If the authors are suggesting that
this is something that could be developed as a corollary or followup of this work, they could clarify
that.

**Author’ response:**

We appreciate the reviewer's comments. We intended to present a positive outlook that future
development of standardized sensor arrays and commercialization through semiconductor processes
will be possible. We think that our combinatorial fabrication strategy through reproducible and precise
automated semiconductor processes presented in this study is advantageous in terms of time and cost
efficiency for extensive investigation, review and database of future optimal material discovery. We
have corrected that sentence (Page 6, Line 191).

“As an applicable future perspective inspired by this study in terms of single gas sensor
research, our design and manufacturing strategy will contribute to overcoming material
wastage and time-consuming trial and error in the process of discovering and optimizing the
optimal combination of sensing materials, which will be expected to be advantageous for
commercialization through mass production.”

**REVIEWER COMMENTS**

**Reviewer #3 (Remarks to the Author):**

1/ Summary of work.

In humans, the olfactory system "senses" a gas, sends message to the central nervous system which
"processes" the information and identifies the gas. This work proposes to solve the gas-sensing
problem with the same framework. Here, the olfactory system is replaced by a sensor array, which
turns gases into some waveform signals. The central nervous system is replaced by a machine
learning model which has two steps, described in lines 299 to 319:

(i) data processing. Preprocess the signals with Fourier transform, take log to get the MFCC
frequency and keep the first 20 terms.

(ii) learn the mapping from (MFCC feature vectors) to (classification of 117 gases). This is a standard
classification problem, and this paper uses a neural network with fixed architecture to solve it.

The authors showed that they achieved between 85% to 99.9% training accuracy depending on the
size of the training set and their training time.

2/ Will the work be of significance to the field and related fields?

I am not well-versed enough in this literature to answer this question. Per the literature cited in the
paper, this approach to build an artificial olfactory system seems new.

3/ Concerns and Flaws.

It seems that existing work have not considered this because reproducibility is a major concern, as,
quoting the paper (lines 55 to 61):

" Even though particular sensing material has high reactivity to a specific gas molecule, the sensitivity
can vary depending on the degree of diffusion of the gas molecule reaching the gas sensor. In
addition, gaseous analytes often exist as complex mixtures of individual gas molecules with varying
concentrations and58

chemical formulas. In this case, since the overall sensitivity can be distorted due to the amplification

or attenuation of the reactivity by the competition between individual gas molecules in the mixed gas,
it
is difficult to expect accurate gas identification results with the sensitivity-oriented data analysis
methods"

The new method, in my understanding, claims to get around this road block as follows.

- 1. Have a standardized sensor array, not one specific one per compound
2. Use machine learning to identify the gases based on their different waveforms.

However, there are two reproducibility issues here:

- 1. Do different arrays reliably produce the same waveform for the same gas.
2. How well does the trained neural network perform out-of-sample on datasets produced by the same
sensor array? How about on datasets produced by a different sensor array? What about datasets with
different mixture ratio of the different gases?

Machine learning systems are notorious for over-fitting, so this is very much a concern. I do not see
an extensive discussion that address these points. The paper uses the word "reproducible" several
1037 times but I do not see data that convinces the reader on this.

**1) Reviewer's comments:**

Will the work be of significance to the field and related fields? I am not well-versed enough in this
literature to answer this question. Per the literature cited in the paper, this approach to build an artificial
olfactory system seems new.

**Author's response:**

We appreciate your interest in our research and your accurate comments. In the field of gas
sensor research, sensitivity (the maximum amplitude of the waveform), reaction time (time to
reach 90% of the peak point of the waveform) and recovery time (time to reach 90% of the
original reference resistance) have been utilized as representative gas sensing characteristics
for identifying gas species. In other words, high sensitivity, fast response and recovery time for

specific gases at low concentrations have been used as key criteria for gas discrimination.
However, existing sensitivity-oriented studies utilizing these magnitude-based parameters have
many fundamental limitations. These parameters are prone to be changed due to changes in the
measurement environment such as temperature and relative humidity as well as measurement
conditions such as volume of the test chamber and location of the sensors. Thus, the cross-
sensitivity due to unknown components may easily occur in an unknown measurement
environment. Data analysis methods based on these magnitude-based parameters are also
difficult to ensure high reliability in terms of reproducibility and stability, making
commercialization and standardization difficult. Above all, the fundamental problem is that
these magnitude-based parameters do not sufficiently reflect the physical and chemical reaction
characteristics between gas molecules and sensing materials. However, there are unique
reaction characteristics between gas molecules and sensing materials, and we can observe them
in the form of time series eigengraphs which we have claimed. Until now, there has been no
interest or effort to fundamentally explore smell itself through the intrinsic attributes of the
eigengraph waveforms according to the unique reaction between gas molecules and reactants.

In other words, based on the analysis principles of eigengraph waveform, it is necessary to
develop new gas sensing characteristics useful for gas identification from wasted data
excluding sensitivity, response, and recovery times. Therefore, in order to generate refined
eigengraphs, it is necessary to develop robust gas sensors that can induce stabilized and
optimized reactions between gas molecules and the sensing material through a uniform and
reproducible manufacturing process with minimal human errors. Thus, Fourier transform-
based MFCC feature vectors were introduced as new gas sensing characteristics to characterize
the eigengraph waveforms according to the unique reaction of gas molecules and sensing
materials. The Fourier transform is very useful when applied to our system based on the
principles of waveform analysis because it can convert any time series signals with similar
amplitudes into frequency signals in which harmonics and fundamental frequencies are
combined at a unique ratio by considering the waveform. Additionally, by further converting
to MFCC, which was developed to mimic human audible characteristics, only the more
important features can be extracted from the FFT spectrum, which makes deep learning
computational efficiency efficient. We have added to the text the importance of utilizing MFCC
feature vectors in the field of gas sensors. (Page 5, Line 131)

“The Fourier transformation is mathematically expressed as an infinite series of products of

the complex exponential functions and their weights (magnitudes of periodic function), which
can be decomposed into the complex number of the sinusoidal periodic by Euler's formula³⁵.
That is, the key value of utilizing the Fourier transform is that any time-domain signal can be
transformed into the frequency-domain components which represent as a unique linear
combination of frequency components consisting of sine and cosine functions with various
magnitude and phase. Accordingly, Fourier transform has been widely applied in various fields
requiring signal processing such as communication engineering, vibration analysis, and speech
recognition^{36, 37}. Therefore, the Fourier transform can be an attractive method to explore the
potential intrinsic attributes of gas sensing properties from eigengraphs with various
waveforms generated by the unique redox reactions between sensing materials and gas
molecules. MFCC are feature vectors developed by imitating the human audible frequency
band caused by the nonlinear structure of the cochlea to characterize the speech signal, which
can be used as a dimensionality reduction method that selects only core features representing
eigengraphs from Fourier transformed frequency components. Finally, in the deep learning
stage, the MFCC feature vectors were input into the deep neural network model which learned
them to classify the identities of mixed gas molecules.”

**Reference (Page 25, Line 826)**

35. Bracewell, R. N. *The Fourier Transform and Its Applications 3rd Edition* (McGraw-Hill,
2000).

36. Weinstein, S. & Ebert, P. Data Transmission by Frequency-Division Multiplexing Using
the Discrete Fourier Transform. *IEEE Trans. Commun.* **19**, 628–634 (1971).

37. Al-Badour, F., Sunar, M. & Cheded, L. Vibration analysis of rotating machinery using
time–frequency analysis and wavelet techniques. *Mech. Syst. Signal Process* **6**, 2083–2101
(2011).

The effectiveness of the MFCC feature vector was verified by distinguishing complex gas
mixtures using a basic DNN architecture. **Consequently, it is meaningful because a series of**
**optimized processes from sensor production, measurement, and data analysis to obtain**
**eigengraphs and MFCC feature vectors explains the data-centric approach.**

**2) Reviewer's comments:**

Do different arrays reliably produce the same waveform for the same gas.

**Author's response:**

We appreciate the reviewer's comments. We fabricated the sensor array using an electron beam
evaporator based on physical vapor deposition. Due to the nature of physical vapor deposition,
the target particles evaporated by the electron beam are deposited uniformly and isotropically,
ensuring excellent film quality over a large area, which makes it possible to reproducibly
manufacture a sensor array that generates uniform data¹. In addition, electron beam deposition
is performed as an automated process under high vacuum conditions while minimizing human
error, making it advantageous for commercialization and standardization based on uniform and
reproducible deposition quality.

To address this, we present uniform data measured on various gas species using four different
gas sensor arrays manufactured through the same process (Supplementary Fig. 4 (Page 7, Line
127), 9(Page12, Line 204). During the measurement process, we used a dedicated measurement
jig capable of measuring four sensor arrays simultaneously (Supplementary Fig. 7a (Page 13,
Line 177)). Additionally, we also present the results of calculating the Euclidean distance in
order to analyze the uniformity between data generated from different samples. Since
presenting numerous tables related to Euclidean similarity analysis may cause fatigue to
reviewers due to limited space, we present a few representative results here. Although it is
cumbersome to explain everything, please refer to the supplementary information
(Supplementary Table 6 (Page 20, Line326), 7(Page 37, Line 439)). The Euclidean distance
values for each channel are shown in a table, and it can be seen that the value of the main
diagonal component is generally the minimum. This means that data between the same
channels were generated uniformly and reproducibly, resulting from the fact that the reaction
characteristics between the gas molecules and the sensing material were maintained stably by
the optimized sensor array manufactured through semiconductor deposition process.

**Reference**

1. Kang M. et al. High Accuracy Real-time Multi-Gas Identification by Batch-Uniform Gas
Sensor Array and Deep Learning Algorithm. *ACS sensors* 7, 430–440 (2022).

**Supplementary Figure 4**

**Supplementary Fig. 4. The database based on the three indistinguishable gas molecules for**
 **deep learning analysis.** Long-term response curves to **a**, Acetone, **b**, Toluene, **c**, Xylene, **d**, AT11, **e**,
 AT13, **f**, AT31, **g**, AX11, **h**, AX13, **i**, AX31, **j**, TX11, **k**, TX13, **l**, TX31 and **m**, ATX111. The sum of
 concentrations of each gas molecule constituting the mixed gases was 2, 4, 6, 8, and 10 ppm. (A:
 acetone, T: toluene, X: xylene, the latter numbers of 11, 13, 31 and 111 mean the mixing ratios of the
 constituent gas molecules).

**Supplementary Fig. 10 | A total of 4 sets of time series eigengraphs generated from the different**
 **4 ORSA chips simultaneously fabricated in same manufacturing process. a, CO. b, CO₂. c, NO.**
 **d, NO₂. e, Gasoline exhaust gas (including CO(5%), CO₂(14%), C₃H₈(0.2%)). f, Diesel exhaust**
 **gas(including CO(5%), CO₂(14%), C₃H₈(0.2%), NO(0.2%), O₂(1%)). g, CO+CO₂. h, CO+CO₂+NO.**

Supplementary Table 6. Euclidean distance for similarity analysis between all
 measurement data of intra-class for the ATX singular and mixed gases.

**Acetone**

A 2ppm	CH1	CH2	CH3	CH4	CH5	CH6	CH7	CH8	CH9
CH1	6,794	18,615	17,093	19,460	16,118	23,725	18,517	21,236	24,027
CH2	18,615	7,018	9,004	8,945	11,079	11,699	10,109	9,732	12,005
CH3	17,093	9,004	1,445	3,361	3,840	7,307	3,276	4,938	7,619
CH4	19,460	8,945	3,361	1,822	5,318	4,952	2,853	2,664	5,508
CH5	16,118	11,079	3,840	5,318	2,131	7,865	3,159	5,632	8,859
CH6	23,725	11,699	7,307	4,952	7,865	380	6,200	3,447	995
CH7	18,517	10,109	3,276	2,853	3,159	6,200	996	2,978	6,262
CH8	21,236	9,732	4,938	2,664	5,632	3,447	2,978	505	3,942
CH9	24,027	12,005	7,619	5,508	8,859	995	6,262	3,942	310

A 4ppm	CH1	CH2	CH3	CH4	CH5	CH6	CH7	CH8	CH9
CH1	10,631	40,817	37,435	41,557	37,015	50,619	41,247	46,831	52,329
CH2	40,817	5,653	11,338	10,825	14,958	14,918	12,891	11,443	16,628
CH3	37,435	11,338	1,568	5,790	6,140	14,960	7,335	11,266	16,671
CH4	41,557	10,825	5,790	1,892	5,554	9,829	3,859	5,927	11,545
CH5	37,015	14,958	6,140	5,554	2,052	14,139	4,864	10,038	15,778
CH6	50,619	14,918	14,960	9,829	14,139	260	10,844	6,369	2,155
CH7	41,247	12,891	7,335	3,859	4,864	10,844	1,155	5,673	12,167
CH8	46,831	11,443	11,266	5,927	10,038	6,369	5,673	925	7,521
CH9	52,329	16,628	16,671	11,545	15,778	2,155	12,167	7,521	389

A 6ppm	CH1	CH2	CH3	CH4	CH5	CH6	CH7	CH8	CH9
CH1	6,103	60,799	57,205	60,400	53,782	72,553	60,244	68,144	75,886
CH2	60,799	4,901	15,446	12,008	18,329	17,783	14,302	13,443	21,116
CH3	57,205	15,446	2,066	9,472	10,855	22,555	12,652	14,302	25,889
CH4	60,400	12,008	9,472	2,564	8,594	13,831	4,858	8,944	16,841
CH5	53,782	18,329	10,855	8,594	1,035	19,598	6,636	14,445	22,326
CH6	72,553	17,783	22,555	13,831	19,598	359	14,853	8,991	3,695
CH7	60,244	14,302	12,652	4,858	6,636	14,853	1,694	5,673	12,167
CH8	68,144	13,443	14,302	8,944	14,445	8,991	5,673	1,453	10,543
CH9	75,886	21,116	25,889	16,841	22,326	3,695	12,167	10,543	495

A 8ppm	CH1	CH2	CH3	CH4	CH5	CH6	CH7	CH8	CH9
CH1	6,014	83,984	80,762	83,520	77,155	97,178	84,137	92,962	101,703
CH2	83,984	4,387	21,263	15,671	21,224	21,361	17,003	17,181	25,884
CH3	80,762	21,263	2,452	16,923	17,708	31,543	21,053	27,402	36,067
CH4	83,520	15,671	16,923	2,248	11,227	18,765	8,479	13,769	22,562
CH5	77,155	21,224	17,708	11,227	1,247	25,357	11,103	19,899	28,710
CH6	97,178	21,361	31,543	18,765	25,357	493	17,611	11,384	5,936
CH7	84,137	17,003	21,053	8,479	11,103	17,611	1,529	11,840	21,500
CH8	92,962	17,181	27,402	13,769	19,899	11,384	11,840	1,962	13,730
CH9	101,703	25,884	36,067	22,562	28,710	5,936	21,500	13,730	560

A 10ppm	CH1	CH2	CH3	CH4	CH5	CH6	CH7	CH8	CH9
CH1	7,767	99,994	95,059	99,196	91,651	117,896	100,666	17,518	124,662
CH2	99,994	4,201	24,640	14,895	21,782	22,729	16,453	17,529	44,858
CH3	95,059	24,640	3,768	18,340	18,577	38,064	23,875	32,926	44,829
CH4	99,196	14,895	18,340	2,491	10,019	21,393	7,217	14,828	26,738
CH5	91,651	21,782	18,577	10,019	2,108	28,619	9,271	21,137	33,335
CH6	117,896	22,729	38,064	21,393	28,619	874	22,383	13,984	7,043
CH7	100,666	16,453	23,875	7,217	9,271	22,383	1,887	12,038	25,399
CH8	17,518	17,529	32,926	14,828	21,137	13,984	12,038	2,628	16,247
CH9	124,662	44,858	44,829	26,738	33,335	7,043	25,399	16,247	677

**Supplementary Table 7. Euclidean distance for similarity analysis between all**
 **measurement data of intra-class for the exhaust gases and their individual**
 **components.**

NO	CH1	CH2	CH3	CH4	CH5	CH6	CH7	CH8	CH9
CH1	9,226	184,444	180,122	113,617	152,902	104,064	145,644	52,714	57,173
CH2	184,444	2,553	8,611	72,066	32,553	81,513	39,378	145,208	241,154
CH3	180,122	8,611	4,354	67,359	27,948	77,021	34,851	140,880	236,851
CH4	113,617	72,066	67,359	6,867	41,457	14,531	34,056	82,639	170,147
CH5	152,902	32,553	27,948	41,457	4,990	50,373	14,195	113,897	209,452
CH6	104,064	81,513	77,021	14,531	50,373	7,054	43,437	71,066	160,191
CH7	145,644	39,378	34,851	34,056	14,195	43,437	3,961	110,460	202,281
CH8	52,714	145,208	140,880	82,639	113,897	71,066	110,460	6,538	102,768
CH9	57,173	241,154	236,851	170,147	209,452	160,191	202,281	102,768	12,208

NO ₂	CH1	CH2	CH3	CH4	CH5	CH6	CH7	CH8	CH9
CH1	709,152	16,600,531	20,158,253	17,152,995	16,020,773	12,901,260	15,793,178	9,606,147	10,155,534
CH2	16,600,531	169,106	3,557,752	653,872	762,849	3,708,264	834,445	7,025,231	6,466,291
CH3	20,158,253	3,557,752	64,447	3,005,524	4,140,204	7,263,827	4,366,048	10,582,922	10,023,941
CH4	17,152,995	653,872	3,005,524	189,132	1,223,447	4,258,774	1,426,126	7,577,767	7,019,176
CH5	16,020,773	762,849	4,140,204	1,223,447	243,883	3,126,900	714,695	6,445,677	5,887,538
CH6	12,901,260	3,708,264	7,263,827	4,258,774	3,126,900	465,892	2,971,793	3,328,363	2,950,715
CH7	15,793,178	834,445	4,366,048	1,426,126	714,695	2,971,793	244,476	6,218,002	5,837,987
CH8	9,606,147	7,025,231	10,582,922	7,577,767	6,445,677	3,328,363	6,218,002	509,387	902,735
CH9	10,155,534	6,466,291	10,023,941	7,019,176	5,887,538	2,950,715	5,837,987	902,735	404,143

CO	CH1	CH2	CH3	CH4	CH5	CH6	CH7	CH8	CH9
CH1	5,716	6,850	10,953	9,667	7,870	10,256	8,255	7,425	7,137
CH2	6,850	4,908	10,785	8,746	7,499	9,511	8,036	7,318	7,409
CH3	10,953	10,785	6,531	10,723	12,039	11,812	8,152	15,686	15,645
CH4	9,667	8,746	10,723	8,697	8,613	8,780	8,368	11,602	12,210
CH5	7,870	7,499	12,039	8,613	6,523	8,835	8,812	7,965	8,921
CH6	10,256	9,511	11,812	8,780	8,835	9,283	9,245	11,862	12,655
CH7	8,255	8,036	8,152	8,368	8,812	9,245	6,998	11,795	11,817
CH8	7,425	7,318	15,686	11,602	7,965	11,862	11,795	4,412	4,690
CH9	7,137	7,409	15,645	12,210	8,921	12,655	11,817	4,690	4,310

CO ₂	CH1	CH2	CH3	CH4	CH5	CH6	CH7	CH8	CH9
CH1	7,025	9,193	10,234	10,961	11,028	8,188	9,518	8,292	7,847
CH2	9,193	2,705	2,579	3,456	3,810	4,737	3,016	4,457	3,889
CH3	10,234	2,579	2,022	2,755	3,333	5,228	2,654	5,148	4,474
CH4	10,961	3,456	2,755	2,883	3,434	6,347	3,277	5,591	5,205
CH5	11,028	3,810	3,333	3,434	3,749	6,523	3,873	5,621	5,443
CH6	8,188	4,737	5,228	6,347	6,523	687	5,663	4,729	5,348
CH7	9,518	3,016	2,654	3,277	3,873	5,663	3,187	5,069	4,337
CH8	8,292	4,457	5,148	5,591	5,621	4,729	5,069	4,523	4,667
CH9	7,847	3,889	4,474	5,205	5,443	5,348	4,337	4,667	4,382

Diesel	CH1	CH2	CH3	CH4	CH5	CH6	CH7	CH8	CH9
CH1	48,302	526,662	448,497	251,766	110,511	199,476	352,028	369,712	374,140
CH2	526,662	5,953	78,181	276,954	580,597	345,156	175,251	161,913	170,269
CH3	448,497	78,181	11,118	47,737	147,468	93,218	56,066	122,793	218,725
CH4	251,766	276,954	198,959	30,765	104,432	58,813	42,950	107,021	175,718
CH5	110,511	580,597	502,577	312,086	55,262	54,445	145,451	32,352	73,631
CH6	199,476	345,156	266,983	74,649	273,834	43,955	97,665	50,988	125,711
CH7	352,028	175,251	97,434	106,299	408,104	170,682	20,345	143,071	216,502
CH8	369,712	161,913	125,038	175,918	422,282	207,516	118,770	23,111	98,139
CH9	374,140	170,269	145,734	183,352	425,878	212,403	129,335	41,263	19,199

Gasoline	CH1	CH2	CH3	CH4	CH5	CH6	CH7	CH8	CH9
CH1	10,323	121,295	161,043	143,335	46,144	87,435	179,978	40,047	61,776
CH2	121,295	14,280	135,115	88,886	78,586	84,757	104,169	87,701	141,643
CH3	161,043	135,115	13,851	47,737	147,468	93,218	56,066	122,793	218,725
CH4	143,335	88,886	47,737	12,228	104,432	58,813	42,950	107,021	175,718
CH5	46,144	78,586	147,468	104,432	10,340	54,445	145,451	32,352	73,631
CH6	87,435	84,757	93,218	58,813	54,445	10,628	97,665	50,988	125,711
CH7	179,978	104,169	56,066	42,950	145,451	97,665	15,721	143,071	216,502
CH8	40,047	87,701	122,793	107,021	32,352	50,988	143,071	7,736	98,139
CH9	61,776	141,643	218,725	175,718	73,631	125,711	216,502	98,139	7,688

CO+CO ₂	CH1	CH2	CH3	CH4	CH5	CH6	CH7	CH8	CH9
CH1	5,942	9,900	12,706	9,372	5,817	7,348	6,835	6,629	7,685
CH2	9,900	3,082	4,557	3,283	8,347	4,999	5,470	6,437	13,748
CH3	12,706	4,557	2,106	5,310	11,157	7,612	7,702	9,188	16,631
CH4	9,372	3,283	5,310	3,556	7,879	4,819	5,263	6,049	13,153
CH5	5,817	8,347	11,157	7,879	5,362	5,943	5,827	5,483	8,347
CH6	7,348	4,999	7,612	4,819	5,943	4,427	4,655	4,840	10,692
CH7	6,835	5,470	7,702	5,263	5,827	4,655	4,183	5,209	10,231
CH8	6,629	6,437	9,188	6,049	5,483	4,840	5,209	4,819	9,562
CH9	7,685	13,748	16,631	13,153	8,347	10,692	10,231	9,562	7,420

CO+CO ₂ +NO	CH1	CH2	CH3	CH4	CH5	CH6	CH7	CH8	CH9
CH1	5,942	9,900	12,706	9,372	5,817	7,348	6,835	6,629	7,685
CH2	9,900	3,082	4,557	3,283	8,347	4,999	5,470	6,437	13,748
CH3	12,706	4,557	2,106	5,310	11,157	7,612	7,702	9,188	16,631
CH4	9,372	3,283	5,310	3,556	7,879	4,819	5,263	6,049	13,153
CH5	5,817	8,347	11,157	7,879	5,362	5,943	5,827	5,483	8,347
CH6	7,348	4,999	7,612	4,819	5,943	4,427	4,655	4,840	10,692
CH7	6,835	5,470	7,702	5,263	5,827	4,655	4,183	5,209	10,231
CH8	6,629	6,437	9,188	6,049	5,483	4,840	5,209	4,819	9,562
CH9	7,685	13,748	16,631	13,153	8,347	10,692	10,231	9,562	7,420

**Supplementary Figure 7a**

**3) Reviewer's comments:**

How well does the trained neural network perform out-of-sample on datasets produced by the
same sensor array? How about on datasets produced by a different sensor array? What about
datasets with different mixture ratio of the different gases?

Machine learning systems are notorious for over-fitting, so this is very much a concern. I do not see
an extensive discussion that address these points. The paper uses the word "reproducible" several
1189 times but I do not see data that convinces the reader on this.

**Author's response:**

We appreciate the reviewer's detailed and accurate comments. Since our system is based on the
principles of waveform analysis using refined eigengraphs, we acknowledge that concerns
about overfitting problems may arise. **We would like to resolve the concerns by conducting**
**additional experiments that take the reviewer's comments and questions into account.** To
address this, we conducted empirical experiments on automobile exhaust using four different
sensor arrays which were fabricated through the same manufacturing process, and performed
4-fold cross-validation based on various deep learning architectures (DNN, CNN, CNN-
LSTM) using the four sets of measured data. As the subject of empirical testing, standard
exhaust gases certified by the KOLAS (Korean Laboratory Accreditation Scheme) were used
for ensuring the precise data measurements and collection (Supplementary Fig. 9 (Page 12,
Line 204)). First, we performed deep learning analysis on both gasoline and diesel exhaust
gases and their individual components (CO, CO₂, NO, NO₂). Second, additional analysis was
performed on the two exhaust gases and another two mixed gases (CO+CO₂, CO+CO₂+NO).
Through the experiment, we can clearly see the differences in the unique graph waveforms for
various mixed gases with different components and different mixing ratios, and based on the
results, confirm that the mixed gases can be completely distinguished. We present the
conclusions from our experiments sequentially below.

First, our sensor arrays which were optimized and functionalized with bimetallic
nanocatalysts through a combinatorial method of noble metals and transition metals, responded
sensitively to mixed gases of various components and single gases, generating eigengraphs
with different waveforms. We present the results of the measurement of them in the manuscript

and supplementary information (Fig. 5 (Page 17, Line 476) and Supplementary Fig. 10 (Page
13, Line 213)). From the results, we clearly demonstrated our arguments that eigengraphs exist
according to the unique reaction characteristics between the sensing materials and gas
molecules.

Second, we confirmed that data generated from different sensor arrays were uniform and
reproducible. We can confirm this through the supplementary fig.10, and the Euclidean
similarity analysis supports these results. In addition, it can be seen that sensor arrays with high
uniformity and reproducibility were fabricated through the electron beam deposition process
we designed, and the optimization strategy stably generated eigengraphs. This is a key element
to contribute to commercialization and standardization.

Third, we extracted MFCC feature vectors to characterize the waveform from the refined
eigengraph, and their applicability was confirmed by testing them on various deep learning
architectures. In the first experiment, we achieved test accuracies of 96.1%, 99.8% and 100%
for DNN, CNN, and CNN-LSTM architectures (Fig. 6 (Page 18, Line 481)) , respectively
(Table 1 (Page 20, Line 549)). Similarly, in the second experiment, we achieved test accuracies
of 99.3%, 99.7% and 99.9%, respectively (Table 2 (Page 20, Line 549)). The effectiveness,
versatility, and expandability of the MFCC feature vectors for various deep learning
architectures were sufficiently proved.

Fourth, in detail, we confirmed that uniformly high accuracy was obtained in each test on
four different data sets with four different samples. Additionally, the training accuracy and loss
function monotonically converge, showing that the learning process was performed well
(Supplementary Fig. 11 (Page 14, Line 218)). From the results, it is confirmed that overfitting
did not occur from deep learning analysis through different datasets of different samples.

Finally, by clearly distinguishing exhaust gases composed of complex gas mixtures, we were
able to confirm the applicability of our data-driven artificial olfactory system to the engineering
model. In follow-up work, we plan to further expand the group of materials capable of
generating unique graphs, develop a variety of optimal sensor arrays, and collect data. In
addition, we will build a data center to store the collected data, develop and produce portable
analysis devices, and identify data about unknown environments by linking them with data
from the data center.

We added a new section to describe the experimental results (Page 16, Line 446).

[revised manuscript text omitted]

Supplementary Fig. 9 | Certified reference materials for the automobile exhaust gases that have
 been manufactured and verified through metrologically valid and strict procedures by KOLAS
 (Korean Laboratory Accreditation Scheme). a, b Standard exhaust gases for legally mandated
 emissions testing for diesel engines including NO (0.18%), CO (4.98%), CO₂ (14.0%), C₃H₈ (0.20%)
 and O₂ (1 %) c, Standard exhaust gases for legally mandated emissions testing for gasoline engines
 including CO (5.01%), CO₂ (14.0%) and C₃H₈ (0.20%). The accompanying standard material certificates
 guarantee the content and relative expanded uncertainty of each component for precise measurements.

Supplementary Figure 11

**Supplementary Fig. 11 | Training result curves of the 4-fold cross validation for the empirical**
 **experiments of the exhaust gases using the three deep learning architecture (CNN, DNN and**
 **CNN-LSTM). Training accuracy and training loss graphs using a, DNN, b, CNN and c, CNN-LSTM for**
 **the two exhaust gases and four individual components. Training accuracy and training loss graphs using**
 **d, DNN, e, CNN and f, CNN-LSTM for four mixed gases including CO+CO₂, CO+CO₂+NO, gasoline**
 **and diesel exhaust gases.**

**Table 1. Results of the 4-fold cross validation using the DNN, CNN and CNN-**
 **LSTM architectures for 2 exhaust gases and 4 individual components.**

4-fold cross validation for 6 gases		Set 1	Set 2	Set 3	Set 4	Average
DNN	Accuracy	96.3 %	95.8 %	95.8 %	96.3 %	96.1 %
	Loss	0.11	0.16	0.10	0.11	0.12
	Training Time	5 min 3 sec	2 min 46 sec	5 min 12 sec	6 min 7 sec	4 min 47 sec
CNN	Accuracy	100 %	99.5 %	99.5 %	100 %	99.8%
	Loss	0.019	0.035	0.061	0.032	0.037
	Training Time	1.1 sec	1.1 sec	0.8 sec	1.2 sec	1.0 sec
CNN-LSTM	Accuracy	100 %	100 %	100 %	100 %	100%
	Loss	0.0074	0.0085	0.0081	0.0087	0.0082
	Training Time	1.8 sec	3.1 sec	1.9 sec	2.0 sec	2.2 sec

**Table 2. Results of the 4-fold cross validation using the DNN, CNN and CNN-**
 **LSTM architectures for 4 mixture gases.**

4-fold cross validation for 4 mixture gases		Set 1	Set 2	Set 3	Set 4	Average
DNN	Accuracy	99.3 %	99.3 %	99.3 %	99.3 %	99.3 %
	Loss	0.023	0.023	0.094	0.022	0.041
	Training Time	25.5 sec	25.8 sec	1 min 7 sec	22.8 sec	35.3 sec
CNN	Accuracy	99.3 %	100 %	99.3 %	100 %	99.7 %
	Loss	0.041	0.064	0.054	0.028	0.047
	Training Time	0.9 sec	0.8 sec	1.3 sec	0.9 sec	1.0 sec
CNN-LSTM	Accuracy	99.3 %	100 %	100 %	100 %	99.9 %
	Loss	0.016	0.0016	0.0016	0.0026	0.013
	Training Time	1.6 sec	1.6 sec	1.6 sec	1.6 sec	1.6 sec

REVIEWERS' COMMENTS

Reviewer #1 (Remarks to the Author):

I am glad that the authors effectively addressed my concerns and challenges in their research work. The authors' ability to provide timely and satisfactory responses to my queries reflects their strong commitment to adhering to scientific principles and conducting reliable research. This dedication benefits the scientific community and enhances our understanding of the subject matter.

Therefore, based on the authors' satisfactory response, I find this version of the article to be acceptable.

Reviewer #2 (Remarks to the Author):

I am largely satisfied with the extensive revisions carried out by the authors.

In particular, I appreciate the new sections on eigengraphs, and on the experiments with automobile exhaust gases.

That said, there are a couple of minor points, I would still like the authors to address before recommending publication.

1. For the first point, it might help to start by quoting the authors, specifically from the eigengraphs section.

"Therefore, based on the above-178 mentioned major four conditions for time-series gas response waveforms, we focused on 179 creating eigengraphs with refined waveforms by controlling our advanced nanotechnology and 180 measurement environments. Eventually, we experimentally found that the eigengraphs can be 181 created by the reaction of specific sensing material and gas molecules using optimized sensor 182 arrays which have 9 different and stable gas sensing characteristics. We formally called this 183 Eigengraphs."

Unless there is some special significance to eigengraphs with the initial 'e' capitalized, the authors seem to be defining them in a circular fashion here. This has to be addressed in a logical way.

On a related note, since this section has been added to develop a "formal definition of the eigengraph and the condition for generating eigengraphs", there should be at least a few-word reflection of this in the abstract - where it appears without introduction.

2. The authors have emphasized a 'standardization' vision, far more now than in the first version. I suggest moderating it because of the following concern.

Hardware specific sensor standardization seems doable from this work, and it can be practically useful; however, standardizing the hardware itself - as suggested by the authors - is not, since that would chill the development of future potentially superior hardware. In fact, ubiquitous standardization of olfactory sensing should perhaps operate with hardware agnostic standards.

**REVIEWER COMMENTS**

**Reviewer #1 (Remarks to the Author):**

I am glad that the authors effectively addressed my concerns and challenges in their research work.

The authors' ability to provide timely and satisfactory responses to my queries reflects their strong

commitment to adhering to scientific principles and conducting reliable research. This dedication

benefits the scientific community and enhances our understanding of the subject matter.

Therefore, based on the authors' satisfactory response, I find this version of the article to be

acceptable.

**Author's response:**

We really appreciate the reviewer's positive comments and encouragement about our works.

**Reviewer #2 (Remarks to the Author):**

I am largely satisfied with the extensive revisions carried out by the authors.

In particular, I appreciate the new sections on eigengraphs, and on the experiments with automobile
exhaust gases.

That said, there are a couple of minor points, I would still like the authors to address before
recommending publication.

1. For the first point, it might help to start by quoting the authors, specifically from the eigengraphs
section.

"Therefore, based on the above-mentioned major four conditions for time-series gas response
waveforms, we focused on creating eigengraphs with refined waveforms by controlling our advanced
nanotechnology and measurement environments. Eventually, we experimentally found that the
eigengraphs can be created by the reaction of specific sensing material and gas molecules using
optimized sensor arrays which have 9 different and stable gas sensing characteristics. We formally
called this Eigengraphs."

Unless there is some special significance to eigengraphs with the initial 'e' capitalized, the authors
seem to be defining them in a circular fashion here. This has to be addressed in a logical way.

On a related note, since this section has been added to develop a "formal definition of the eigengraph
and the condition for generating eigengraphs", there should be at least a few-word reflection of this in
the abstract - where it appears without introduction.

2. The authors have emphasized a 'standardization' vision, far more now than in the first version. I
suggest moderating it because of the following concern.

Hardware specific sensor standardization seems doable from this work, and it can be practically
useful; however, standardizing the hardware itself - as suggested by the authors - is not, since
that 1496 would chill the development of future potentially superior hardware. In fact, ubiquitous

standardization of olfactory sensing should perhaps operate with hardware agnostic
standards. 1498

**1) Reviewer's Comments:**

[1] For the first point, it might help to start by quoting the authors, specifically from the eigengraphs
section.

"Therefore, based on the above-mentioned major four conditions for time-series gas response
waveforms, we focused on creating eigengraphs with refined waveforms by controlling our advanced
nanotechnology and measurement environments. Eventually, we experimentally found that the
eigengraphs can be created by the reaction of specific sensing material and gas molecules using
optimized sensor arrays which have 9 different and stable gas sensing characteristics. We formally
called this Eigengraphs."

[2] Unless there is some special significance to eigengraphs with the initial 'e' capitalized, the authors
seem to be defining them in a circular fashion here. This has to be addressed in a
logical way. 1512

[3] On a related note, since this section has been added to develop a "formal definition of the
eigengraph and the condition for generating eigengraphs", there should be at least a few-word
reflection of this in the abstract - where it appears without introduction.

**Author's response:**

We sincerely appreciate the reviewer's careful and thoughtful comments and efforts to improve
the quality of our manuscript. We acknowledge that there were ambiguities in our description
about the eigengraphs section.

[1] We appreciate the reviewer's helpful suggestions. The relevant content about the major
four 1522 conditions for time-series gas reaction waveforms that we mentioned in the introduction
section 1523 has been linked to this paragraph as follows.

"Therefore, based on the major four conditions for time-series gas response waveforms described in the
introduction section, we focused on finding optimized signals with refined waveforms by controlling
our advanced nanotechnology and measurement environments.

[2] We completely agree with what the reviewer said, and it seems that the term 'eigengraphs'
itself is also occasionally used circularly in other academic fields. However, we think its 1530
significance has not yet received attention in the field of some nanoscience and nanotechnology
that accompanies with electrochemical reactions. In the field of electrochemistry, the results of
electron transfer resulting from chemical reactions are generally observed in the form of
electrical signals. In this study, to implicatively represent the results of naturally existing 1534
intrinsic electrochemical reactions between nanomaterials, we introduced the term 'Eigengraph' 1535
which combines the word 'graph' and the prefix 'Eigen' which has 'unique' properties and 1536
meaning. In addition, the term 'Eigengraphs' with the initial 'e' capitalized was intended to be 1537
used as a keyword to highlight our findings and formalize the use of this term in the field of
1538 electrochemistry.

Based on the reviewer's opinion, we have added additional explanations to support the
logicity for the use of the initial 'e' capitalized of the eigengraphs. Additionally, the terms
'eigengraph' used before the sentence formalizing our findings on 'the law of existence of 1542
Eigengraph in electrochemistry' have been replaced by another appropriate terms to smoothen 1543
the logical development of our arguments.

“Therefore, based on the major four conditions for time-series gas response waveforms described in the
introduction section, we focused on finding optimized signals with refined waveforms by controlling
our advanced nanotechnology and measurement environments. Eventually, we experimentally found
that different intrinsic signals can be created by the redox reactions between specific sensing materials
and gas molecule using an optimized sensor array which have 9 independent and stable gas sensing
characteristics. Thus, we formalize the findings as the law of existence of Eigengraph in
electrochemistry, which is intended to represent the existence of natural intrinsic electrochemical
reactions among the nanomaterials. In order for the eigengraphs to be commonly used with engineering
credibility, researchers must create an internationally standardized and optimized device and
measurement environment. Based on this, it is necessary to find and collect the eigengraphs that satisfy
the major four conditions of time-series gas response data for various gas molecules under various
environmental conditions to identify unknown odors. The law of existence of Eigengraph in
electrochemistry can be applicable to all fields dealing with various electrochemical reactions and
provide potential opportunities to inherently understand their unique characteristics by extracting
significant attributes from eigengraphs. Consequently, it will ultimately be possible to predict unknown
target molecules through the eigengraphs for well-established reactions based on advanced

nanotechnology.”

[3] We have reflected the contents of the "The law of existence of Eigengraph in electrochemistry and
its implementation conditions" in the abstract. And we revised by shortening the abstract to 150 words.

“Recent studies of electronic nose system tend to waste significant amount of important data in odor
identification. Until now, the sensitivity-oriented data composition has made it difficult to discover
meaningful data to apply artificial intelligence in terms of in-depth analysis for odor attributes
specifying the identities of gas molecules, ultimately resulting in hindering the advancement of the
artificial olfactory technology. Here, we realize novel data-centric approaches to implement
standardized artificial olfactory systems inspired by human olfactory mechanisms by formally defining
and utilizing the concept of Eigengraph in electrochemisty. The implicit odor attributes of the
eigengraphs were mathematically substantialized as the Fourier transform-based Mel-Frequency
Cepstral Coefficient feature vectors. Their effectiveness and applicability in deep learning processes for
gas classification have been clearly demonstrated through experiments on complex mixed gases and
automobile exhaust gases. We suggest that our findings can be widely applied as source technologies
to develop standardized artificial olfactory systems.”

**2) Reviewer’s Comments:**

The authors have emphasized a 'standardization' vision, far more now than in the first version. I
suggest moderating it because of the following concern.

Hardware specific sensor standardization seems doable from this work, and it can be practically
useful; however, standardizing the hardware itself - as suggested by the authors - is not, since that
would chill the development of future potentially superior hardware. In fact, ubiquitous
standardization of olfactory sensing should perhaps operate with hardware agnostic standards.

**Author’s response:**

We fully empathize with reviewer's concerns about cooling down in development of potentially
superior hardware in the future that our proposed vision of standardizing devices will cause. 1589
We hope that our research results will contribute to opening up infinite amount of creativities 1590 and
possibilities for developing various superior hardware in the future, including standardized

sensors for various material groups. Therefore, we have moderated our vision of 1592
standardization towards the hardware itself and revised our manuscript by removing terms
related to the standardization of hardware.

[1] Page 12, Line 333

“In order for the eigengraphs to be commonly used with engineering credibility, researchers must create
an internationally standardized and optimized gas sensors and appropriate peripheral hardware that
ensures reliability in various measurement environment.”

[2] Page 20, Line 549

“Most importantly, we emphasize that international discussions and continuous efforts for the 1600
development of adequate devices and instruments are needed to explore and secure standard
eigengraphs for various odors.”
